# Selective endocytosis controls slit diaphragm maintenance and dynamics in *Drosophila* nephrocytes

Konrad Lang[1], Julian Milosavljevic[1], Helena Heinkele[1], Mengmeng Chen[1], Lea Gerstner[1], Dominik Spitz[1], Severine Kayser[1], Martin Helmstädter[1], Gerd Walz[1,2], Michael Köttgen[1,2], Andrew Spracklen[3†], John Poulton[4], Tobias Hermle[1]*

[1]Renal Division, Department of Medicine, Faculty of Medicine and Medical Center - University of Freiburg, Freiburg, Germany; [2]CIBSS – Centre for Integrative Biological Signalling Studies, Freiburg, Germany; [3]Lineberger Comprehensive Cancer Center, University of North Carolina at Chapel Hill, Chapel Hill, United States; [4]Department of Medicine, Division of Nephrology and Hypertension, University of North Carolina at Chapel Hill, Chapel Hill, United States

**Abstract** The kidneys generate about 180 l of primary urine per day by filtration of plasma. An essential part of the filtration barrier is the slit diaphragm, a multiprotein complex containing nephrin as major component. Filter dysfunction typically manifests with proteinuria and mutations in endocytosis regulating genes were discovered as causes of proteinuria. However, it is unclear how endocytosis regulates the slit diaphragm and how the filtration barrier is maintained without either protein leakage or filter clogging. Here, we study nephrin dynamics in podocyte-like nephrocytes of *Drosophila* and show that selective endocytosis either by dynamin- or flotillin-mediated pathways regulates a stable yet highly dynamic architecture. Short-term manipulation of endocytic functions indicates that dynamin-mediated endocytosis of ectopic nephrin restricts slit diaphragm formation spatially while flotillin-mediated turnover of nephrin within the slit diaphragm is needed to maintain filter permeability by shedding of molecules bound to nephrin in endosomes. Since slit diaphragms cannot be studied in vitro and are poorly accessible in mouse models, this is the first analysis of their dynamics within the slit diaphragm multiprotein complex. Identification of the mechanisms of slit diaphragm maintenance will help to develop novel therapies for proteinuric renal diseases that are frequently limited to symptomatic treatment.

## Editor's evaluation

This article would be of interest to all researchers who work in understanding the mechanisms involved in podocyte slit diaphragm homeostasis and maintenance of the glomerular filtration barrier. The work provides substantial new insights into nephrin dynamics and the mechanisms of slit diaphragm maintenance. A series of compelling experiments depicted that dynamin-mediated endocytosis was involved in ectopic nephrin turnover and that flotillin-mediated turnover of nephrin occurred within the slit diaphragm was needed to maintain filter permeability in-vivo.

## Introduction

The human kidneys maintain water and electrolyte homeostasis and efficiently excrete metabolic waste products and xenobiotics. The essential first step of kidney function is to generate primary

*For correspondence: tobias.hermle@uniklinik-freiburg.de

Present address: †Department of Biology, University of Massachusetts, Amherst, United States

Competing interest: The authors declare that no competing interests exist.

urine by filtration of blood across a size- and charge-selective filter. Every single day, the kidneys are perfused with ~1700 l of blood and filter about 180 l of nearly protein-free of primary urine – thus retaining approximately 12 kg of plasma protein from the filtered fraction. It remains unclear, how it is possible to maintain this filter during constant filtration without leakage of plasma protein or clogging while adapting to changing physiological conditions (*Scott and Quaggin, 2015*; *Butt et al., 2020*).

The filtration barrier is provided by two epithelial layers, the vascular endothelium and the glomerular podocytes with their interjacent basement membrane. The filtrate traverses through endothelial pores, the basement membrane, and narrow filtration slits that form between the elaborate network of the podocytes' interdigitating foot processes (*Scott and Quaggin, 2015*). These slits are guarded by the slit diaphragm whose major structural components are nephrin and NEPH1 that engage transcellularly (*Kestilä et al., 1998*; *Holzman et al., 1999*; *Donoviel et al., 2001*; *Gerke et al., 2003*; *Barletta et al., 2003*). However, slit diaphragms represent a multiprotein complex that includes further proteins such as podocin (*Grahammer et al., 2013*; *Boute et al., 2000*) and associates with proteins like TRPC6 (*Winn et al., 2005*; *Reiser et al., 2005*) to direct signaling (*Martin and Jones, 2018*; *Grahammer et al., 2016*). Several lines of evidence support a role of endocytic pathways for proper function of the filtration barrier (*Inoue and Ishibe, 2015*). Overexpressed nephrin is subject to endocytosis in vitro (*Quack et al., 2006*; *Qin et al., 2009*) and dysregulation of endocytosis in murine podocytes resulted in severe proteinuria, the clinical hallmark of a failing glomerular filter (*Harris et al., 2011*; *Bechtel et al., 2013*; *Soda et al., 2012*). In mice, PKC-α and CIN85 promote nephrin endocytosis under diabetic conditions and similarly after angiotensin II exposure (*Tossidou et al., 2010*; *Teng et al., 2016*; *Quack et al., 2011*; *Königshausen et al., 2016*). We and others discovered monogenic mutations of endosomal regulators as the molecular cause of severe proteinuria in humans (*Hermle et al., 2018*; *Dorval et al., 2019*; *Kampf et al., 2019*). Surprisingly, mutations of these widely expressed genes exclusively manifested with nephrotic syndrome (*Hermle et al., 2018*; *Dorval et al., 2019*; *Kampf et al., 2019*). While endocytosis occurs ceaselessly in all cells, the kidney's filtration barrier thus requires a particularly tight regulation of endocytic trafficking. Endocytosis might be needed for slit diaphragm formation, renewal, and/or restriction of slit diaphragms to their proper location. Elucidating these fundamental aspects of podocyte biology represents a major challenge due to the rapid transport dynamics and complex architecture of the filtration barrier. Overexpression of nephrin in immortalized cells rendered significant insights, but the lack of slit diaphragms in in vitro models entails that nephrin is not embedded within a proper multiprotein complex. Genetic mouse models allowed identification of essential genes but they cannot provide insights into dynamic remodeling/recycling of the slit diaphragm due to slow throughput and limited accessibility. Thus, we employed the podocyte-like nephrocytes in *Drosophila* that form functional slit diaphragms using orthologous proteins (*Denholm and Skaer, 2009*; *Zhuang et al., 2009*; *Hermle et al., 2017*; *Helmstädter et al., 2012*). Utilizing this model, we developed assays to examine slit diaphragm dynamics directly after short-term manipulation of endocytic functions and obtained unique in vivo insights into the filtration barrier's dynamics. Lateral diffusion of ectopic nephrin is prevented by rapid dynamin-dependent endocytosis restricting slit diaphragm localization. In contrast, nephrin engaged within the proper slit diaphragm complex is constantly endocytosed flotillin2-dependently followed by recycling. Such turnover offers flexibility and cleanses the filtration barrier from adherent molecules, maintaining its permeability. Selective and functionally distinct routes of endocytic transport thus maintain barrier's architecture and permeability.

## Results

### Slit diaphragms in nephrocytes are stable structures that dynamically reconstitute upon disruption

The podocyte-like nephrocytes form functional slit diaphragms that filter larval plasma before entry into membrane invaginations termed labyrinthine channels (schematic *Figure 1A*). The *Drosophila* slit diaphragm proteins Sns and Kirre are orthologous to human nephrin and NEPH1 respectively and engage in a multiprotein complex (*Weavers et al., 2009*; *Hochapfel et al., 2017*; *Hermle et al., 2017*). For simplicity, we will use the human names for the *Drosophila* orthologs throughout the manuscript. As a consequence of the nephrocyte's cytoarchitecture, these proteins stain in a linear pattern reminiscent of fingerprints in tangential sections and adhere to the cell membrane in a dotted line in

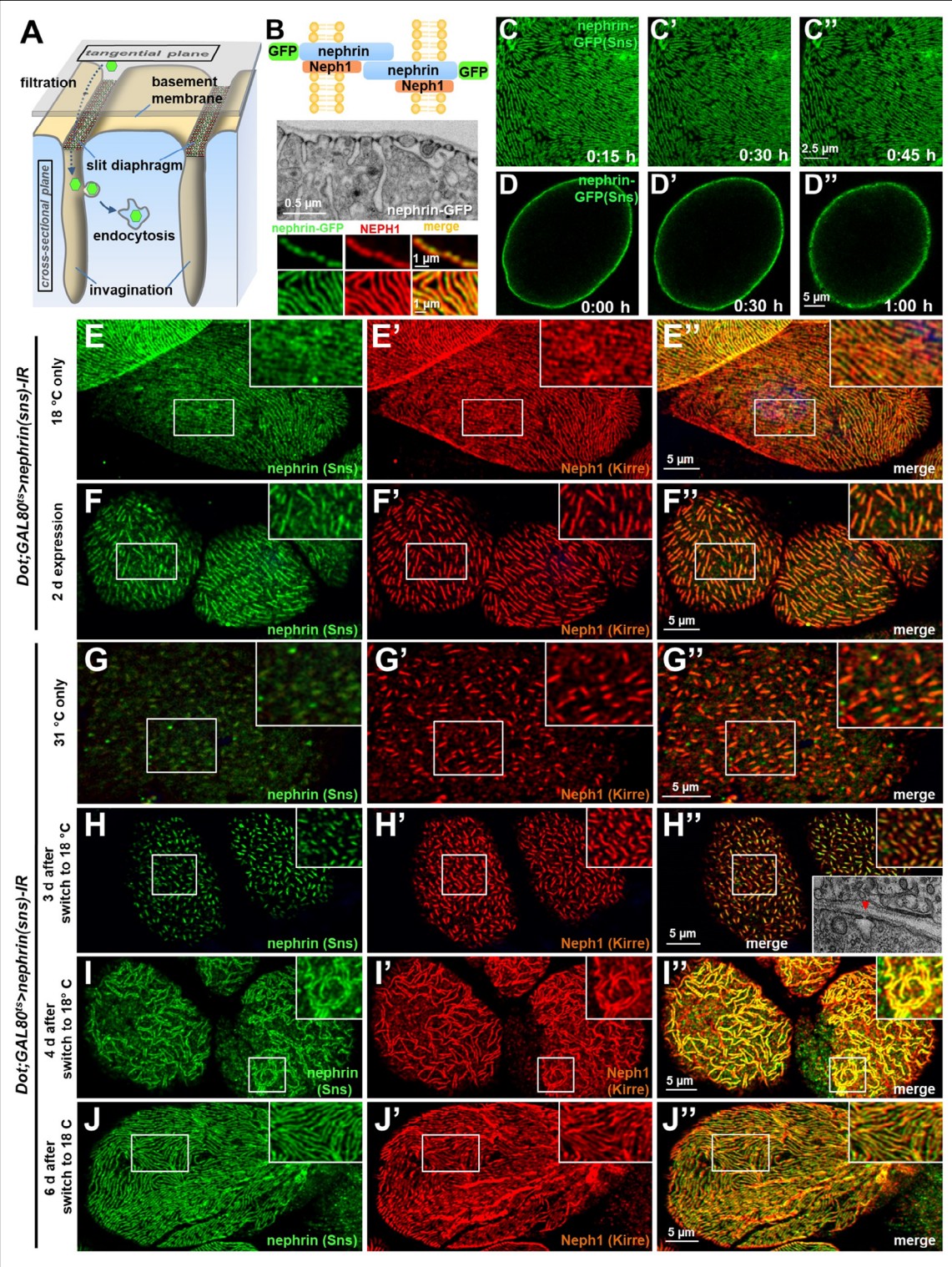

**Figure 1.** Slit diaphragms proteins form a stable architecture that is re-established upon disruption. (**A**) Schematic illustrating the nephrocyte ultrastructure and function (surface detail). Molecules destined for removal (shown as green hexagons) pass a bi-layered filtration barrier before being subject to endocytosis within membrane invaginations. (**B**) The schematic (upper section) illustrates the slit diaphragm after knock-in of GFP into the nephrin locus. The transmission electron microscopy image (middle section) shows a surface detail of a nephrocyte expressing nephrin-GFP homozygously with regular slit diaphragms. Confocal images (lower section) of a nephrin-GFP nephrocyte show colocalization with endogenous Neph1 (Kirre) in cross-sectional (upper row) and tangential sections (lower row). (**C–D″**) Snapshots from a movie obtained by live-cell imaging reveal a stable slit diaphragm pattern in the tangential section (**C–C″**). This is confirmed by cross-sectional analysis in the same genotype (**D–D″**) where no vesicles for

*Figure 1 continued on next page*

*Figure 1 continued*

bulk transport of nephrin are observed. (**E–F"**) Confocal images of tangential section of nephrocytes stained for slit diaphragm proteins while silencing of fly nephrin (*sns*) is blocked by *GAL80*^ts at 18°C show a regular staining pattern (**E–E"**). A temperature shift to 31°C initiates RNAi expression, resulting in reduction of approximately 50% of the slit diaphragm protein after 2 days (**F–F"**). (**G–I"**) Confocal images of tangential section of nephrocytes that express nephrin (sns)-RNAi and *GAL80*^ts continuously at a non-inhibiting temperature of 31°C stained for slit diaphragm proteins nephrin (sns) and Neph1 (Kirre) show an extensive loss of nephrin staining after silencing while a punctate pattern of Neph1 (lacking its binding partner) is observed (see also magnified inset) (**G–G"**). Both proteins colocalize in short lines indicating renewed formation of slit diaphragms after a temperature shift to 18°C that inhibits RNAi expression for 3 days (**H–H"**). Inset in (**H"**) shows transmission electron microscopy of the same stage with return of sparse and isolated slit diaphragms (red arrowhead). The longer lines of slit diaphragm proteins begin to cluster in pairs or triplets after another day, covering a large part of the cell surface in a wide-meshed network (**I–I"**). (**J–J"**) Slit diaphragm architecture is restored after blocking the expression of nephrin-RNAi for 6 days.

The online version of this article includes the following figure supplement(s) for figure 1:

**Figure supplement 1.** Validation of nephrin-GFP and additional time points for disruption and reassembly of slit diaphragms.

cross sections (*Hermle et al., 2017*; *Figure 1A–B*, *Figure 1—figure supplement 1A-A"*). To explore slit diaphragm dynamics, we introduced GFP into the C-terminus of the endogenous *nephrin* (*sns*) locus via genome editing (schematic of slit diaphragm with tag *Figure 1B*). Genomic nephrin-GFP resulted in expression of functional protein that sustained regular slit diaphragms in a homozygous state (*Figure 1B* electron micrograph). Nephrin-GFP colocalized with endogenous Neph1 (*Figure 1B* bottom panels, *Figure 1—figure supplement 1B-C*), suggesting integration into the slit diaphragm. Confirming its specificity, the GFP-derived fluorescence was abrogated by nephrin silencing (*Figure 1—figure supplement 1D-D"*). Using this model, we studied slit diaphragm dynamics by live-cell imaging ex vivo. We observed a stable slit diaphragm architecture over a period of up to 1 hr (*Figure 1C–D*, *Videos 1–2*). To explore the half-life of wild-type nephrin, we employed a temperature-sensitve *GAL80* to modify *GAL4*-dependent transgene expression by temperature shifts (active at 31°C, inactive at 18°C). Upon short-term expression of *nephrin*-RNAi, we observed an incremental loss of nephrin protein, as indicated by shorter slit diaphragm lines. A reduction of approximately 50% in length and density was reached after *nephrin* silencing for 2 days (compare *Figure 1E–E"* and *Figure 1F–F"*, silencing for 24 hr *Figure 1—figure supplement 1E-E"*). This implies an extensive half-life for nephrin protein ranging from ~1 to 3 days. To investigate if slit diaphragms may reconstitute after disruption, we used *GAL80*^ts/*GAL4* to first silence nephrin (*Figure 1G–G"*) before stopping RNAi expression, which resulted in slow return of nephrin after 3–4 days. Few isolated slit diaphragms were detectable in electron microscopy (EM) at that

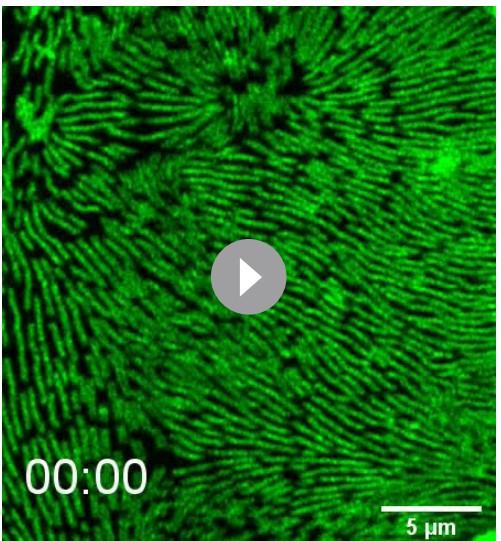

**Video 1.** Nephrin-GFP in larval nephrocyte tangential section. A movie obtained by confocal live-cell imaging reveals a stable slit diaphragm pattern in the tangential section. Time stamp indicates elapsed time in minutes.
https://elifesciences.org/articles/79037/figures#video1

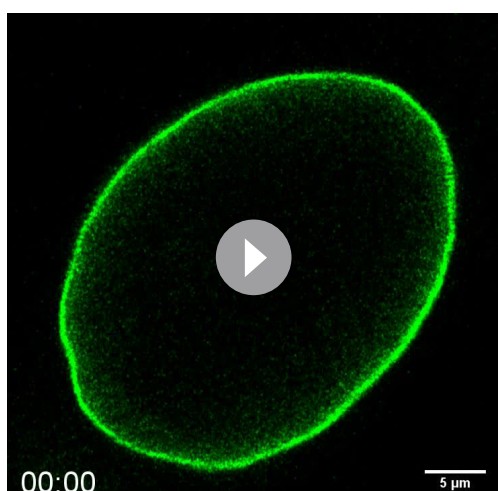

**Video 2.** Nephrin-GFP in larval nephrocyte cross section. Movie obtained by confocal live-cell imaging is shown. No vesicles for bulk transport of nephrin are observed during the observation peroid. Time stamp indicates elapsed time in minutes.
https://elifesciences.org/articles/79037/figures#video2

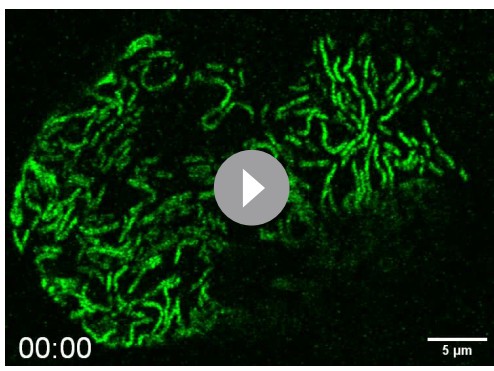

**Video 3.** Nephrin-GFP in larval nephrocyte upon transient nephrin silencing. Confocal live-cell imaging of nephrin-GFP nephrocytes 3.5 days after transient silencing of nephrin reveals that reconstitution of slit diaphragms occurs slowly with minor changes during the observation period after ceased RNAi expression. Time stamp indicates elapsed time in minutes.
https://elifesciences.org/articles/79037/figures#video3

stage (inset *Figure 1H"*). In confocal microscopy the lines of slit diaphragms elongated over time (*Figure 1H*, additional images *Figure 1—figure supplement 1G-J"*), and gradually repopulated the nephrocyte's surface, frequently in pairs (*Figure 1I*, live-cell imaging *Figure 1—figure supplement 1J-J"*, *Video 3*), and finally restored normal density (*Figure 1J–J"*). Slit diaphragms thus are formed by a protein with an extensive half-life and may reconstitute after disruption.

## Live antibody labeling and FRAP suggest rapid slit diaphragm turnover

We hypothesized that slit diaphragms are subject to endocytic turnover in vivo as previously suggested by in vitro studies (*Inoue and Ishibe, 2015*). To study the dynamics of nephrin within the slit diaphragm, we introduced a Myc-tag into the extracellular domain of nephrin by CRISPR-editing the second exon (*Figure 2—figure supplement 1A*). Myc-staining revealed a typical slit diaphragm pattern, colocalizing with Neph1 (*Figure 2A*, *Figure 2—figure supplement 1B-C"*). The Myc-signal was lost upon nephrin silencing (*Figure 2—figure supplement 1D-D"*), and homozygous animals formed regular slit diaphragms (*Figure 2B*). This indicates that a functional fusion protein is expressed from the edited locus. The extracellular tag was labeled ex vivo by exposing living nephrocytes to anti-Myc antibody. We tracked the fate of the live antibody-labeled nephrin protein by further incubating the living cells (chase period). After fixation and permeabilization, regular Myc-staining was employed to detect the entire nephrin protein (schematic *Figure 2C*). Without chasing, the live labeled antibody matched the pattern obtained by the subsequent total stain (*Figure 2D–D"*), which confirms efficient live labeling. With progressive incubation time, the signal from live labeling at the slit diaphragms decreased, while a faint, diffuse intracellular signal appeared (*Figure 2E–F"*). Residual signal of live labeled antibody at the slit diaphragm persisting even after 2 hr suggested a small immobile fraction. At the end of chasing, we further detected a slit diaphragm pattern that was exclusively derived from the total staining while largely lacking in live labeling (*Figure 2F–F"*). This indicates that during the chase period new protein had reached intact slit diaphragms. Quantification of the surface-derived Myc-nephrin signal in ratio to the submembraneous intracellular signal indicated steady reduction over time, supporting a constant endocytic turnover (*Figure 2G*, *Figure 2—figure supplement 1E*). Apparently, the live labeled antibody was rapidly degraded, but we detected a vesicular signal when degradation was slowed by bafilomycin-mediated inhibition of endolysosomal acidification (*Figure 2—figure supplement 1F-G*). To evaluate nephrin turnover independently, we employed the CRISPR-edited nephrin-GFP to perform fluorescence recovery after photobleaching (FRAP) experiments (*Figure 2H*, quantitation *Figure 2I*). Confirming our findings with live antibody labeling, FRAP analysis indicated a rapid recovery after bleaching with 50% recovery of the signal after ~7 min. The recovery reached a plateau within 30 min, suggesting an immobile fraction of fly nephrin that is not replaced by turnover during the observation period. However, the majority of nephrin protein undergoes quite rapid cycles of turnover.

## Rab5 regulates trafficking of fly nephrin

We now wanted to explore how manipulation of the endocytic activity affects nephrin. The most basic steps of endocytosis are uptake followed by sorting either toward degradation or recycling to the plasma membrane. The small GTPase Rab5 localizes to early endosomes, where it regulates uptake, endosomal fusion, and cargo sorting (schematic *Figure 3A*). We overexpressed the constitutively active *Rab5^{Q88L}* in nephrocytes limiting expression to 24 hr using *GAL80^{ts}* to avoid non-specific effects. YFP-*Rab5^{Q88L}* induced enlarged endosomes containing nephrin (*Figure 3B–B"*, control *Figure 3—figure*

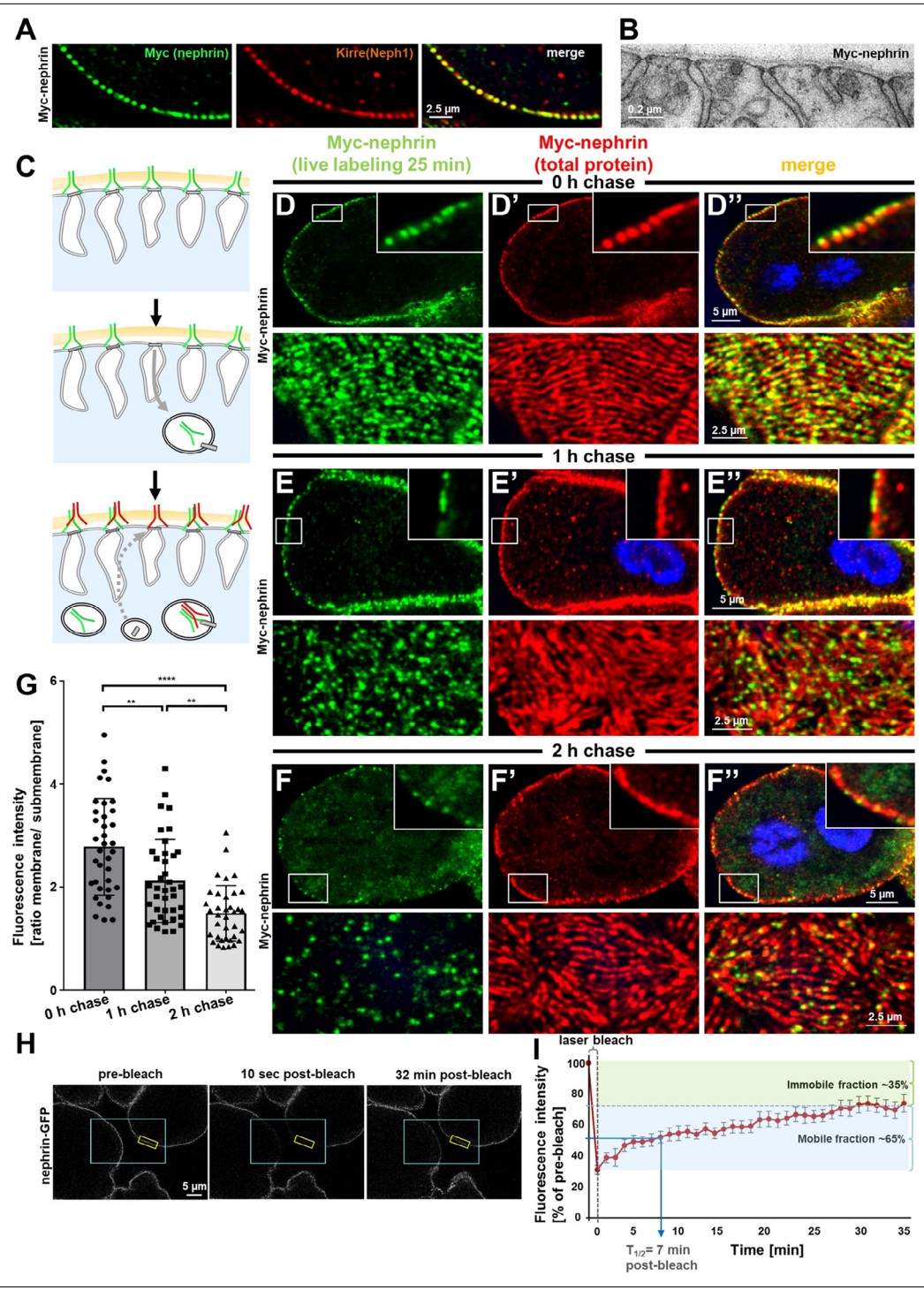

**Figure 2.** Live antibody labeling shows rapid nephrin turnover. (**A**) Immunostaining of nephrocyte expressing Myc-nephrin homozygously shows colocalization with endogenous Neph1. (**B**) Transmission electron microscopy of a nephrocyte expressing Myc-nephrin homozygously reveals regular slit diaphragms suggesting the tagged protein is functional. (**C**) Schematic illustrating live antibody labeling: Living nephrocytes are labeled with anti-Myc antibody (green) that may undergo endocytosis during chasing. Total nephrin stain follows after fixation and permeabilization (red). Colocalization of green and red indicates stable nephrin (surface) or endocytosed nephrin (subcortical). Exclusively green signal indicates antibody dissociation, while new nephrin reaching the surface during the chase period will stain only red. (**D**) Confocal microscopy images show cross sections (top) and tangential sections (bottom) from Myc-nephrin nephrocytes after live antibody labeling without chasing.

*Figure 2 continued on next page*

*Figure 2 continued*

Extensive colocalization indicates successful nephrin labeling. Nuclei are marked by Hoechst 33342 in blue here and throughout the figure. (**E**) Confocal images analogous to (**D**) but after 1 hr of chasing reveal incipient endocytosis. (**F**) Confocal images analogous to (**D–D″**) but after 2 hr of chasing suggest extensive endocytosis. Diffuse intracellular signal from live labeling suggests that internalized antibody separated from nephrin. Exclusively red nephrin signal indicates newly delivered protein. (**G**) Quantitation of fluorescence intensity derived from live labeling from conditions in (**D–F**) expressed as a ratio of surface (slit diaphragm) and subcortical areas confirms significant nephrin turnover (mean ± standard deviation, n=12–13 animals per $p < 0.01$ for chase of 1 hr and $p < 0.0001$ for 2 hr). (**H**) Shown are frames from a time lapse movie of nephrin-GFP nephrocytes. The blue box demarcates the region of photobleaching, the yellow box outlines a region of interest (ROI) where the fluorescence intensity was measured over the length of the fluorescence recovery after photobleaching (FRAP) experiment. A loss of fluorescence intensity compared to pre-bleach condition (left panel) is detectable 10 s after photobleaching (middle panel). After 32 min, the fluorescence recovers significantly (right panel). (**I**) Quantitative analysis from multiple FRAP experiments (n=5 cells, 8 ROIs total, mean ± standard deviation) reveals an initially rapid recovery of fluorescence intensity that slows to a plateau suggesting a nephrin half-life of ~7 min. The majority of nephrin molecules (~65%) are replaced within 30 min (mobile fraction).

The online version of this article includes the following figure supplement(s) for figure 2:

**Figure supplement 1.** Validation of Myc-nephrin and bafilomycin treatment.

---

*supplement 1A-B″*). This indicated that increased Rab5 function redirected nephrin to endosomes. Subsequently, we evaluated short-term silencing of *Rab5* for 17 hr. This time sufficed to extensively reduce Rab5 protein (*Figure 3—figure supplement 1C-D*), without affecting cellular viability since nephrocytes remained negative for cell death marker terminal deoxynucleotidyl transferase dUTP nick end labeling (TUNEL, *Figure 3—figure supplement 1E-E'*, positive control *Figure 3—figure supplement 1F-F'*). In this early phase of disrupted endocytosis, the lines of slit diaphragm proteins became blurry and began to fuse (*Figure 3C*). In cross sections, we observed extensive translocation of nephrin from the cell surface deeper into the cell (*Figure 3D*). After prolonged *Rab5* silencing for 24 hr, we observed a localized breakdown of slit diaphrams on sections of the cell surface (*Figure 3E–F″*). This was matched by gradual expansion of slit diaphragm gaps in live imaging (*Figure 3G–G″*, *Video 4*). To confirm a Rab5-specific effect, we employed dominant negative *Rab5^{S43N}*, which phenocopied our findings using *Rab5*-RNAi (*Figure 3—figure supplement 1G-H″*). Rab5 disruption thus has a severe impact on slit diaphragm maintenance. To correlate the subcortical nephrin with potential aberrant endosomes, we exposed nephrocytes during acute silencing of *Rab5* to an extended course of tracer FITC-albumin which is rapidly endocytosed by nephrocytes (*Hermle et al., 2017*). Despite partial silencing of *Rab5,* we observed significant tracer endocytosis under these conditions (*Figure 3H–H″*). While this observation confirmed preserved cell viability and residual endocytic activity, we did not observe colocalization of the endocytic tracer and subcortical nephrin. Nephrin thus translocates extensively to an ectopic location that differs from an (early) endosomal compartment.

## *Rab5* silencing causes lateral diffusion of slit diaphragm proteins and alters filtration characteristics

We hypothesized that lateral diffusion of slit diaphragm proteins into labyrinthine channels contributes to intracellular translocation of nephrin during acute silencing of *Rab5*. To simultaneously visualize the nephrocytes' labyrinthine channels and nephrin, we filled these invaginations via passive diffusion by incubating nephrocytes in Texas-Red-Dextran (10 kDa) after brief fixation before staining nephrin (*Figure 4A*). This approach reflected normal channel morphology in control cells (*Figure 4B-B″*), as well as the expected loss of the invaginations upon nephrin silencing (*Figure 4—figure supplement 1A-A″*). While nephrin was absent from the membrane invaginations under control conditions (*Figure 4B-B″*), we observed partial colocalization of ectopic nephrin with the channels in *Rab5*-RNAi nephrocytes (*Figure 4C-C″*). This suggested that nephrin partially translocated to the membrane invaginations upon disruption of endocytosis. Live imaging showed increasing formation of clusters of nephrin-GFP below the cell membrane which preceded the localized breakdown (*Figure 4D*, *Video 5*) Live imaging further indicated dynamic movement of subcortical nephrin, likely caused by moving labyrinthine channels. Nephrin was removed in vesicles, suggesting residual, but misdirected endocytosis (*Figure 4E*, *Video 6*). EM uncovered slit diaphragms deeply within the labyrinthine channels

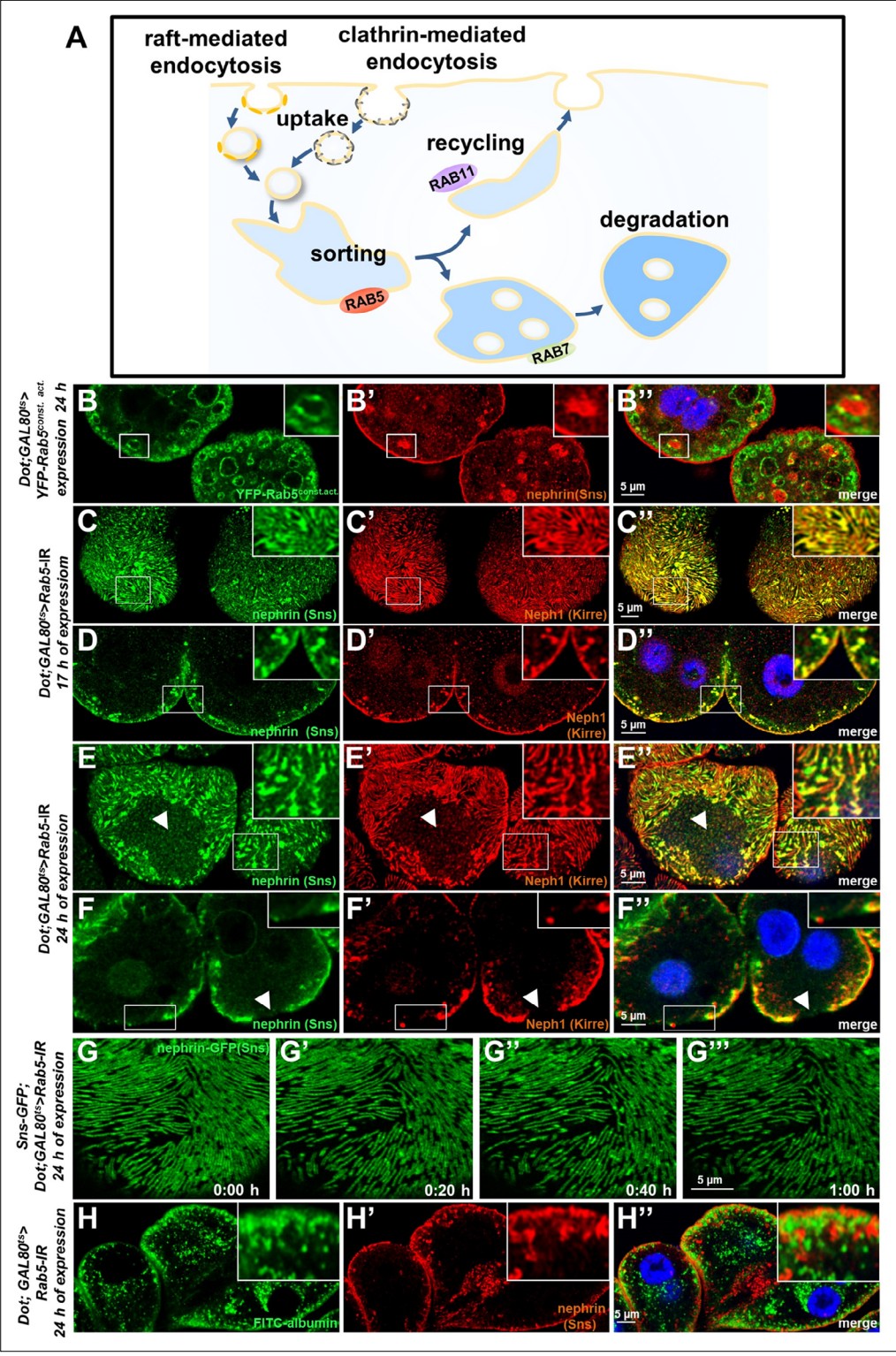

**Figure 3.** Endosomal regulator Rab5 is required for maintenance of slit diaphragms. (**A**) Schematic illustrating endocytic trafficking in a simplified manner shows raft-mediated and clathrin-mediated uptake converging in the early endosome by vesicle fusion. Uptake, early endosome formation and cargo sorting are controlled by Rab5. Sorting may direct cargo either toward degradation, which is promoted by Rab7, or back toward the cell membrane by recycling pathways such as Rab11-dependent recycling. (**B–B″**) Cross-sectional confocal microscopy images from nephrocytes expressing constitutively active YFP-*Rab5* for 24 hr (green) show highly enlarged early endosomes that contain ectopic fly nephrin (see also magnified inset). Nuclei are marked by Hoechst 33342 in

*Figure 3 continued on next page*

*Figure 3 continued*

blue here and throughout the figure. (**C**) Confocal images of nephrocytes with acute silencing of *Rab5* for 17 hr reveals brighter sections within the lines of slit diaphragm protein in tangential sections. Lines further are blurry and focally confluent (see also magnified inset). (**D**) Cross-sectional images of nephrocytes with short-term silencing of *Rab5* show appearance of ectopic slit diaphragm protein below the surface (compare to control *Figure 3—figure supplement 1A-A''*). (**E–F**) Tangential sections (**E**) and cross sections (**F**) of nephrocytes with slightly longer silencing of *Rab5* for 24 hr stained for nephrin (Sns) and Neph1 (Kirre) reveal progressive thickening of slit diaphragms and localized breakdown of the slit diaphragms in a circumscribed area (white arrowheads). (**G–G'''**) Snapshots from a movie obtained by live-cell imaging using confocal microscopy are shown. Nephrocytes expressing nephrin-GFP (heterozygously) are shown after 24 hr of acute *Rab5* silencing. Increasing gaps and a progressive reduction of slit diaphragms are observed over the course of 1 hr. Cells with a mild phenotype were chosen for live-cell imaging to ensure cellular viability. The nephrin signal in tangential sections appears slightly less blurry compared to untagged nephrin. (**H–H''**) Confocal microscopy images showing cross sections of nephrocytes after 24 hr of *Rab5* silencing are shown. Living cells were exposed to FITC-albumin (green) for 15 min before fixation and staining for nephrin (red). Cells show significant endocytosis of FITC-albumin indicating cell viability and residual endocytic activity despite silencing of *Rab5*. Ectopic nephrin and FITC-albumin do not colocalize, indicating that ectopic nephrin is not found within a subcellular compartment that is also destination for recently endocytosed cargo.

The online version of this article includes the following figure supplement(s) for figure 3:

**Figure supplement 1.** Validation and control experiments for loss-of-function of *Rab5*.

often in rosette-like clusters upon acute silencing of *Rab5* (*Figure 4G*, control *Figure 4F*). This further supports lateral diffusion of nephrin protein, likely due to insufficient removal of the ectopic nephrin caused by impaired endocytosis.

To evaluate if *Rab5*-RNAi alters nephrocyte filtration barrier permeability, we recorded simultaneous endocytosis of tracers FITC-albumin (66 kDa) that is close to the filtration barrier's size limit for passage (*Hermle et al., 2017*) and the considerably smaller tracer Texas-Red-Dextran (10 kDa). In nephrocytes expressing *Rab5*-RNAi, the decrease in uptake of FITC-albumin was about twice as strong as reduction of the smaller Texas-Red-Dextran compared to control conditions (*Figure 4H–I''*). In contrast, nephrin silencing reduced uptake of both tracers equally (*Figure 4J–J''*). Accordingly, the ratio of the fluorescence of the small tracer relative to the large tracer was strongly elevated for Rab5, while the ratio was unchanged by nephrin silencing (*Figure 4K*). This observation suggests a reduced permeability of the slit diaphragm for larger tracer following disruption of endocytosis (*Figure 4L*). We simultaneously exposed nephrocytes to another pair of tracers (Texas-Red-Avidin, 66 kDa, and Alexa Fluor 488 wheat germ agglutinin, 38 kDa) and *Rab5*-RNAi in turn affected uptake of the larger tracer more severely (*Figure 4—figure supplement 1B-D*). Comparing the rate of passive diffusion of FITC-albumin and Texas-Red-Dextran (10 kDa) across the slit diaphragm into labyrinthine channels after brief fixation of nephrocytes similarly indicated reduced penetrance of the larger tracer (*Figure 4—figure supplement 1E-G*). Taken together, we conclude that defective endocytosis alters permeability of the nephrocyte's slit diaphragm in a size-dependent manner, suggesting incipient filter clogging.

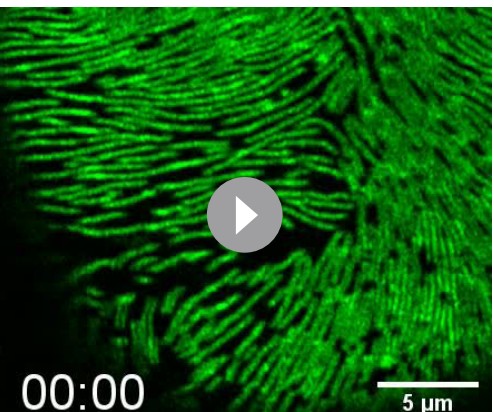

**Video 4.** Nephrin-GFP in larval nephrocyte upon acute Rab5 silencing (surface). Confocal live-cell imaging using is shown. Nephrocytes expressing nephrin-GFP (heterozygously) are recorded after 24 hr of acute Rab5 silencing. Increasing gaps and a progressive reduction of slit diaphragms are observed over the observed period. Cells with a mild phenotype were chosen for live-cell imaging to ensure cellular viability. The nephrin signal in tangential sections appears slightly less blurry compared to endogenous nephrin.

https://elifesciences.org/articles/79037/figures#video4

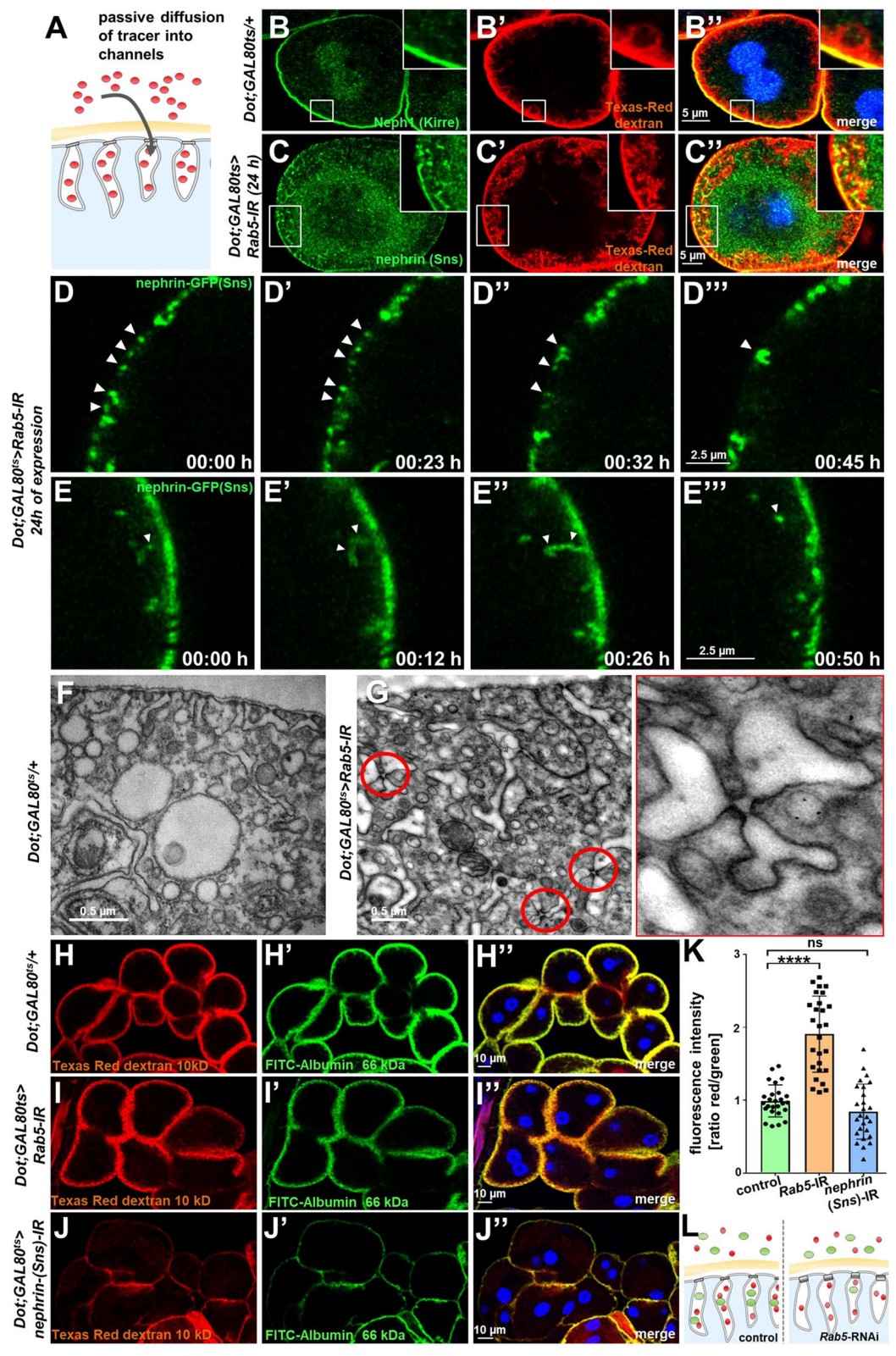

**Figure 4.** Endocytosis prevents lateral diffusion of nephrin and preserves filter permeability. (**A**) Schematic illustrates the assay for visualization of labyrinthine channels. Nephrocytes are fixed briefly before exposure to Texas-Red-Dextran that enters the channels by passive diffusion. (**B**) Confocal microscopy image of a control nephrocyte is stained for Neph1 (green) together with labeling of the channels by Texas-Red-Dextran (10 kDa,

*Figure 4 continued on next page*

*Figure 4 continued*

red). Channels extend directly below the slit diaphragms. Nuclei are marked by Hoechst 33342 in blue here and throughout the figure. (**C**) Confocal images of nephrocytes with short-term silencing of *Rab5* show mislocalized fly nephrin below the cell surface that colocalizes significantly with the labyrinthine channels visualized by Texas-Red-Dextran (10 kDa). (**D–E'''**) Snapshots from movies obtained by live-cell imaging are shown. Nephrocytes express nephrin-GFP (heterozygously) concomitant with *Rab5*-RNAi for 24 hr. Fusion and cluster formation (white arrowheads in panels D) of fly nephrin precedes appearance of gaps (**D–D'''**). Similarly, formation of protrusions of slit diaphragm proteins from the cell surface is followed by a formation of vesicles (**E–E'''**, white arrowheads). (**F**) Electron microscopy (EM) image from a cross section through the surface of a control nephrocyte reveals regular slit diaphragms bridging the membrane invaginations called labyrinthine channels. (**G**) EM image from a section through the surface of a nephrocyte expressing *Rab5*-RNAi acutely for 24 hr demonstrates ectopic formation of slit diaphragms forming rosette-like structures within the labyrinthine channels (red circles, magnification on the right). (**H–J''**) Confocal microscopy images of nephrocytes after simultaneous uptake of tracers FITC-albumin (66 kDa, green) and Texas-Red-Dextran (10 kDa) are shown. Control nephrocytes show robust uptake of both tracers (**H–H''**). Silencing of *Rab5* acutely for 24 hr shows a stronger decrease in the uptake of the larger tracer FITC-albumin compared to smaller Texas-Red-Dextran (**I–I''**). Both tracers are equally reduced upon *nephrin* silencing (**J–J''**). (**K**) Quantitation of fluorescence intensity expressed as a ratio of Texas-Red-Dextran/FITC-albumin (small/large tracer) confirms a disproportionate reduction for the larger tracer for *Rab5*-RNAi but not *nephrin*-RNAi (mean ± standard deviation, n=9 animals per genotype, p<0.0001 for *Rab5*-RNAi, p>0.05 for *nephrin*-RNAi). (**L**) Schematic illustrates how incipient filter clogging affects uptake of larger tracer disproportionately.

The online version of this article includes the following figure supplement(s) for figure 4:

**Figure supplement 1.** Channel diffusion assay reveals loss of invaginations upon silencing of nephrin and impaired slit diaphragm passage upon silencing of *Rab5*.

## Slit diaphragm maintenance requires endocytosis and recycling but not degradation

To explore the contribution of key aspects of endocytic cargo processing, we studied the effect of silencing critical Rab proteins. Expression of *Rab7*-RNAi, directed against the major Rab GTPase promoting degradation, or expression of dominant negative *Rab7* had no overt effect on the slit diaphragm architecture (*Figure 5A-A''*, *Figure 5—figure supplement 1A-A''*, Rab7 staining of control vs. knockdown *Figure 5—figure supplement 1B-C''*). However, *Rab7*-RNAi caused an additional faint nephrin signal in the cell (*Figure 5A–A''*, *Figure 5—figure supplement 1A-A''*, *Figure 5—figure supplement 1C-C''*), suggesting accumulation of endocytosed nephrin protein due to lack of degradation. This further suggests that the vast

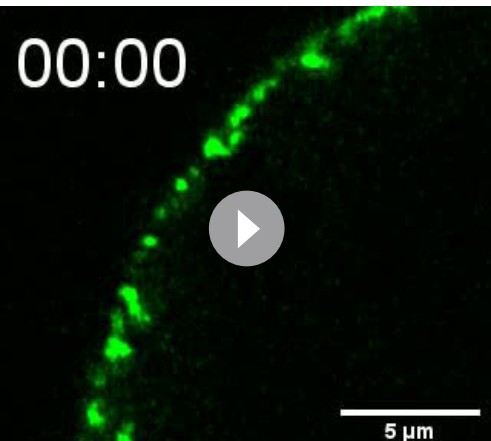

**Video 5.** Nephrin-GFP in larval nephrocyte upon acute Rab5 silencing (breakdown of slit diaphragms). Shown is a movie obtained by confocal live-cell imaging. Nephrocytes recorded in a tangential section express nephrin-GFP (heterozygously) concomitant with *Rab5*-RNAi for 24 hr. Fusion and cluster formation of fly nephrin precedes appearance of growing gaps (D–D'''). https://elifesciences.org/articles/79037/figures#video5

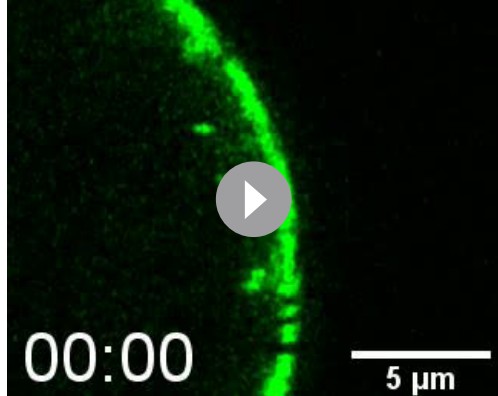

**Video 6.** Nephrin-GFP in larval nephrocyte upon acute Rab5 silencing (lateral diffusion). Confocal live-cell imaging of (heterozygously) nephrin-GFP expressing nephrocyte after 24 hr of *Rab5*-RNAi expression shows formation of protrusions of slit diaphragm protein from the cell surface followed by formation of vesicles. https://elifesciences.org/articles/79037/figures#video6

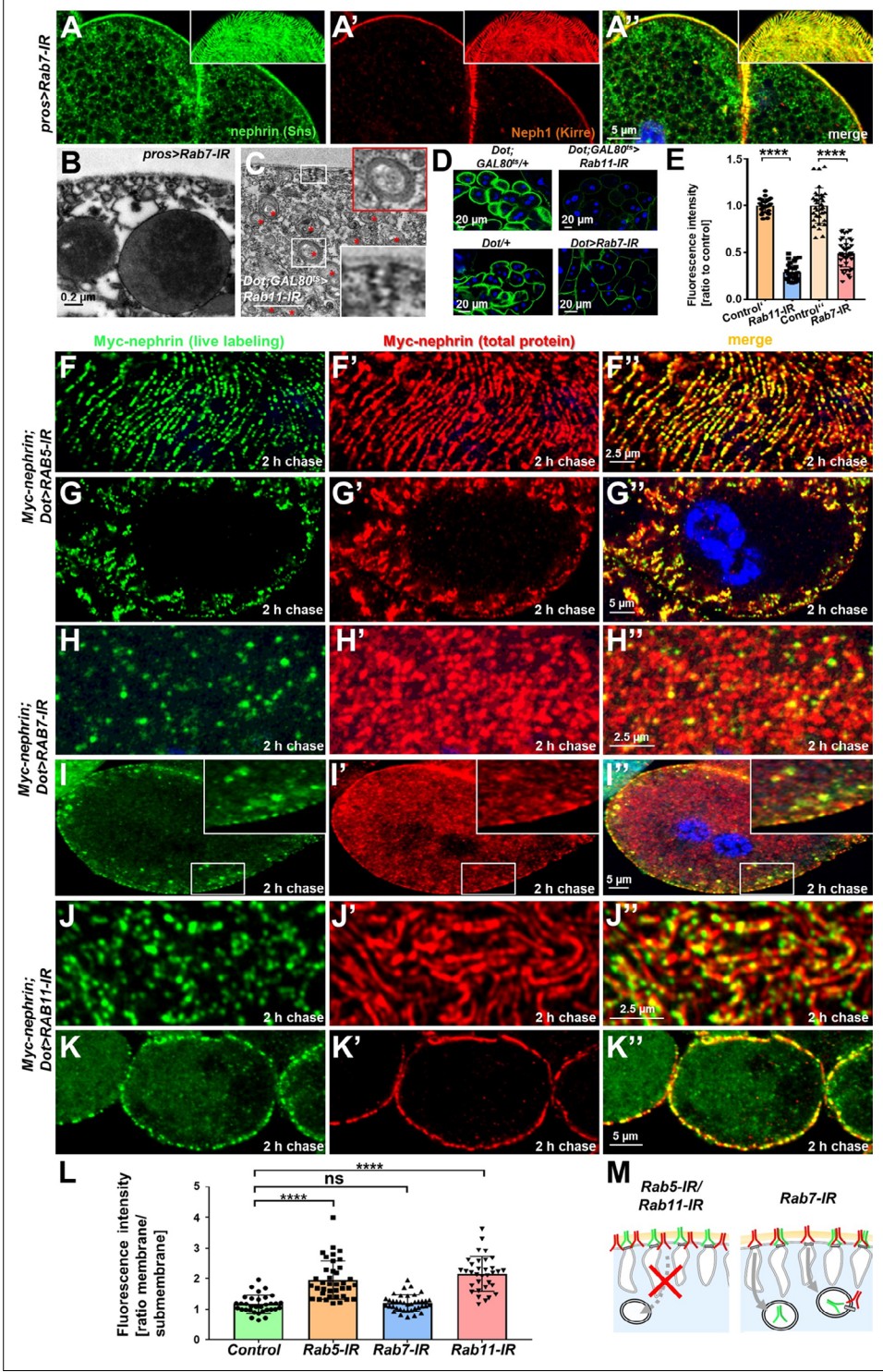

**Figure 5.** Endocytic uptake and Rab11-dependent recycling are required for slit diaphragm maintenance.
(**A–A''**) Stainings of *Rab7*-RNAi nephrocytes reveal an additional faint signal for nephrin but not for Neph1 that
likely reflects accumulation of nephrin upon defective degradation. Tangential sections (insets) show a regular
fingerprint-like pattern, indicating undisturbed slit diaphragm formation. Nuclei are marked by Hoechst 33342 in
blue here and throughout the figure. (**B**) Electron microscopy (EM) image of Rab7-RNAi nephrocyte shows normal
slit diaphragms and large vesicles. (**C**) EM of nephrocyte expressing *Rab11*-RNAi reveals reduction of labyrinthine
channels with multiple slits close to the cell surface (see inset) and expansion of lysosomes (red asterisks, see
also magnified inset). Scale bar represents 0.2 μm. (**D**) FITC-albumin endocytosis as assay for nephrocyte function

*Figure 5 continued on next page*

*Figure 5 continued*

shows reduced uptake for *Rab7*-RNAi (lower panels) and *Rab11*-RNAi (upper panels) using *Dorothy-GAL4* or *prospero*-GAL4 compared to the respective controls. (**E**) Quantitation of results from (**D**) in ratio to a control experiment performed in parallel (mean ± standard deviation, n=11–14 animals per genotype, p<0.0001 for *Rab7*-RNAi and n=9 animals per genotype p<0.0001 for *Rab11*-RNAi). Sidak post hoc analysis was used to correct for multiple comparisons. (**F–K″**) Confocal microscopy images of tangential sections (**F–F″, H–H″, J–J″**) and cross sections (**G–G″, I–I″, K–K″**) of Myc-nephrin nephrocytes after live antibody labeling and 2 hr of chasing are shown for the indicated genotypes. Silencing of *Rab5* at 18°C was obtained before flies were adapted to 25°C for 1 hr (**F-G″**). Live labeling (green) and total stain (red) show near-complete colocalization for *Rab5*-RNAi (**F–G″**), indicating disrupted nephrin turnover. Extensive amounts of subcortical nephrin are revealed in cross sections (**G–G″**), compatible with lateral diffusion into the membrane invaginations. Cells expressing *Rab7*-RNAi after live antibody labeling show undisturbed nephrin turnover as the live labeled antibody is removed from the surface (**H–H″**). Cross sections of *Rab7*-RNAi nephrocytes reveal numerous subcortical vesicles that partially show isolated signal for the live labeling, indicating the antibody disengaged from nephrin (**I–I″**). Nephrocytes expressing *Rab11*-RNAi show strong retention of live labeled nephrin on the cell surface (**J–J″**), suggesting impaired turnover. Cross sections show the antibody on the surface, but not in labyrinthine channels (**K–K″**). (**L**) Quantitation of results from (**F–K″**) expressed as ratio of the fluorescence intensity between surface and subcortical region for individual cells (mean ± standard deviation, n=11–13 animals per genotype, p<0.0001 for *Rab5*-RNAi, p>0.05 for *Rab7*-RNAi and p<0.0001 for *Rab11*-RNAi). (**M**) Schematic illustrates findings studying nephrin live labeling upon silencing of *Rab5/Rab7/Rab11*.

The online version of this article includes the following figure supplement(s) for figure 5:

**Figure supplement 1.** Validation and controls for *Rab7* and *Rab11*.

**Figure supplement 2.** Additional images for live antibody.

---

majority of endocytosed nephrin returns to the plasma membrane through recycling pathways. EM also revealed accumulation of electron-dense vesicles morphologically compatible with accumulating autophagolysosomes (*Spitz et al., 2022*) upon expression of *Rab7*-RNAi (*Figure 5B*). We previously observed progressive loss of nephrin after acute silencing of *Rab11* (*Kampf et al., 2019*). We confirmed a role for recycling using a second *Rab11*-RNAi, observing coarser and wider spaced slit diaphragms (*Figure 5—figure supplement 1D-E''*), with nephrin and Neph1 appearing independently on the cell surface (insets in *Figure 5—figure supplement 1E-E''*, Rab11 staining of control vs. knockdown *Figure 5—figure supplement 1F-G*). Compensatory transport through alternative pathways such as *Rab4*-mediated recycling thus may be less effective in maintaining the slit diaphragm proteins in their stoichiometry and coherence during transport. Ultrastructural analysis upon silencing of *Rab11* revealed formation of multiple slits within shortened labyrinthine channels and further excessive formation of lysosomes (red asterisks, *Figure 5C*). As all steps of endocytosis are connected, this led us to hypothesize that lack of recycling intensifies degradation but also attenuates uptake. Accordingly, we observed reduced FITC-albumin endocytosis following silencing of *Rab11* (*Figure 5D–E*), suggesting that reduced uptake and intensified degradation partially compensate for compromised recycling. We further observed a diminished FITC-albumin endocytosis (*Figure 5D–E*) for *Rab7*-RNAi suggesting that uptake attenuates upon defective degradation as well. We investigated the role of these Rab proteins specifically for nephrin turnover using the live antibody labeling assay. As described above, in control cells, this assay indicated extensive replacement of nephrin at slit diaphragms after 2 hr (control with heterozygous Myc-nephrin *Figure 5—figure supplement 2A-A''*). In contrast, nephrocytes expressing *Rab5*-RNAi at 18°C to attain a milder loss-of-function retained the live labeled antibody after 2 hr at the membrane but also within lines likely corresponding to labyrinthine channels (*Figure 5F–G''*). Removal of live labeled Myc-nephrin thus depends entirely on endocytosis. Subsequently, we carried out the live antibody labeling assay in nephrocytes expressing *Rab7*-RNAi and removal of the Myc-antibody indicated unimpaired endocytic uptake despite *Rab7*-silencing (*Figure 5H–I''*). However, cross sections revealed vesicles containing Myc-antibody (*Figure 5H–I''*). Hence, the decelerated degradation facilitated tracking of the antibody's endocytosis. Interestingly, a majority of vesicles were positive for the live labeled Myc antibody, but negative for the total Myc-Nephrin or nephrin co-staining (*Figure 5I–I''*, *Figure 5—figure supplement 2B-C''*). This indicates that the live labeled antibody and Myc-nephrin had dissociated extensively upon entry into endosomes. This implies a functional role for endocytosis by shedding of unwanted molecules from nephrin suggesting that constant endocytosis facilitates self-cleansing of the filtration barrier. Finally, we evaluated the impact

of silencing of *Rab11*, which had a similar impact on nephrin turnover as expression of *Rab5*-RNAi (*Figure 5J–K*, quantitation *Figure 5L*, schematic *Figure 5M*). However, while overall nephrin turnover was similarly reduced, *Rab11*-RNAi did not cause lateral diffusion of slit diaphragm protein into the labyrinthine channels (compare *Figure 5K–K″* to *Figure 5G–G″*). This suggests that divergent routes of endocytosis are required for nephrin turnover and prevention of lateral diffusion.

## Dynamin-dependent endocytosis and raft-mediated endocytosis play distinct roles in filtration barrier maintenance

*Rab5* orchestrates endocytic sorting downstream of virtually all entry pathways. Since slit diaphragms form in raft domains, nephrin might travel by clathrin- or raft-mediated endocytosis as suggested by findings in vitro (*Qin et al., 2009*). Recently, a role for clathrin-mediated uptake was further suggested by studies in pericardial nephrocytes (*Wang et al., 2021*). To assess the specific role of these uptake pathways for nephrin trafficking, we first inhibited dynamin-mediated endocytosis. This more canonical route of entry includes clathrin-mediated endocytosis. To disrupt dynamin short term, we employed a temperature-sensitive mutant of the *Drosophila* dynamin gene, *shibire*[ts]. This variant remains functional at lower temperatures but a temperature shift effectively blocks dynamin-mediated endocytosis in nephrocytes at 30°C (*Kosaka and Ikeda, 1983*). Nephrocytes were phenotypically normal in animals kept at 18°C (*Figure 6—figure supplement 1A-B″*). Blocking dynamin for 2 hr by shifting the animals to 30°C resulted in a staining pattern of nephrin and Neph1 that phenocopied *Rab5-RNAi* showing lateral diffusion (*Figure 6A–B″*). This suggested that removal of ectopic nephrin requires a dynamin-dependent route of entry. To obtain acute inhibition of raft-mediated endocytosis, we exposed nephrocytes ex vivo to methyl-β-cyclodextrin (cylodextrin) for 2 hr. This compound depletes the plasma membrane of cholesterol which disperses lipid rafts and thereby prevents raft-mediated endocytosis (*Zidovetzki and Levitan, 2007*). However, this short-term treatment had no effect on the staining pattern of slit diaphragm proteins (*Figure 6C–D″*), suggesting that removal of ectopic nephrin exclusively relies on dynamin-mediated endocytosis. In contrast, when we studied nephrin turnover by live labeling, we observed effective clearance of the live labeled Myc-antibody from the slit diaphragms for *shibire*[ts] nephrocytes (*Figure 6E–F″* quantitation *Figure 6I*, Neph1 co-staining *Figure 6—figure supplement 1C-C″*). Hence, slit diaphragm turnover does not require dynamin. However, when we dispersed lipid rafts by cyclodextrin, we observed a strong reduction in nephrin turnover, using the live antibody labeling assay (*Figure 6G–H″*, quantitation *Figure 6I*). This indicates that raft-mediated endocytosis is required for the rapid internalization of nephrin residing within the slit diaphragm. We conclude that selective transport routes regulate free nephrin vs. slit diaphragm-associated nephrin in vivo.

## Flotillin2-mediated endocytosis is required for nephrin turnover in *Drosophila* nephrocytes

We next sought to identify the mediator that promotes raft-dependent endocytosis of nephrin. Since caveolins are absent from the *Drosophila* genome, we hypothesized that flotillins play this role. Flotillins associate with the inner leaflet of the plasma membrane initiating raft-mediated endocytosis in response to phosphorylation by the kinase Fyn (*Glebov et al., 2006*; *Meister and Tikkanen, 2014*; *Otto and Nichols, 2011*). We expressed *flotillin2*-RNAi in nephrocytes and observed impaired nephrocyte function (*Figure 7A–C*). Staining nephrocytes for fly nephrin and Neph1, we observed a localized breakdown of slit diaphrams on sections of the cell surface similar to prolonged *Rab5* silencing with incomplete penetrance (*Figure 7D–E″*). Since some animals showed no overt phenotype similar to short-term cyclodextrin (*Figure 7—figure supplement 1A-B″*), we hypothesized that the localized breakdown of slit diaphragms may only occur as a long-term consequence. Importantly, when we performed the live antibody labeling after silencing *flotillin2*, we detected strongly diminished nephrin turnover (*Figure 7F–G*). This suggests that nephrocytes exercise the specific nephrin turnover by *flotillin2*-dependent endocytosis. Studying size-dependent permeability of slit diaphragms upon silencing of *flo2*, we observed a phenocopy of *Rab5*-RNai with relatively stronger reduction of uptake for the tracer closer to the size cut-off of the nephrocyte filtration barrier that is between 66 and 80 kDa (*Hermle et al., 2017*; *Figure 7H–J*). Another *flo2*-RNAi recapitulated the observed effects on FITC-albumin uptake, slit diaphragm protein stainings, and altered filtration barrier permeability (*Figure 7—figure supplement 1C-J*). This confirms that silencing flottilin-mediated turnover

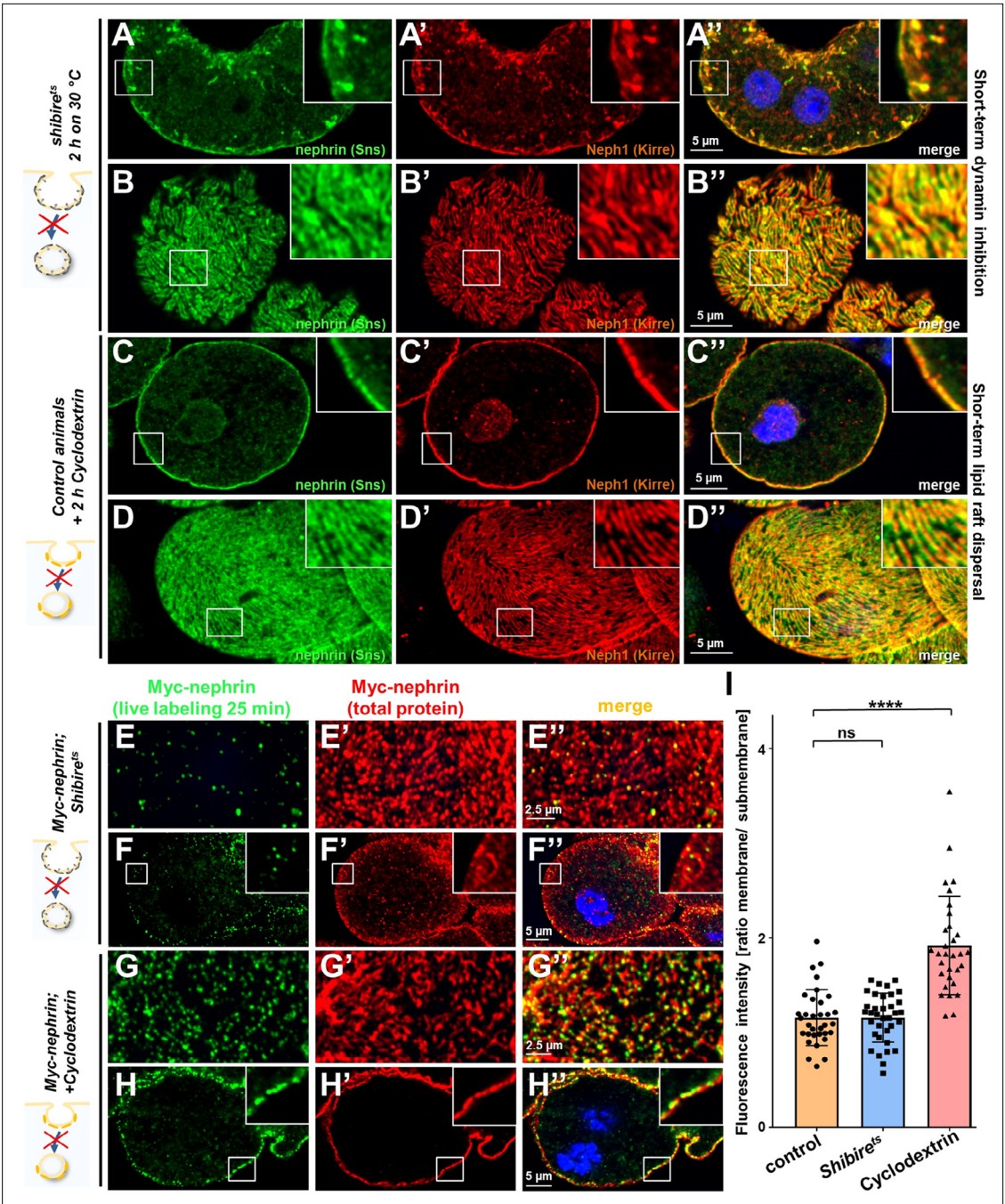

**Figure 6.** Differential transport through dynamin-mediated or raft-mediated endocytosis is required for slit diaphragm maintenance in nephrocytes. (**A–B"**) Confocal images of nephrocytes stained for slit diaphragm proteins carrying a temperature-sensitive variant (*G141S*) of *shibire*, the *Drosophila* dynamin, homozygously. The mutant protein is functional at lower temperatures but lacks function at 30°C and the animals were exposed to 30°C for 2 hr before staining. Cross sections show accumulation of subcortical slit diaphragm protein in clusters and short lines protruding from the surface (**A–A"**). Tangential sections indicate a mild confluence and few brighter clusters of slit diaphragm proteins (**B–B"**). (**C–D"**) Confocal images of control nephrocytes treated with cylodextrin for 2 hr ex vivo to inhibit raft-mediated endocytosis show a regular staining pattern of slit diaphragm proteins in cross-sectional (**C–C"**) and tangential planes (**D–D"**). (**E–H"**) Confocal microscopy images showing tangential sections (panels E and G) and cross sections (panels F and G) of nephrocytes carrying one copy of the genomic Myc-nephrin after live antibody labeling with 2 hr of chase period are for the indicated genotypes or interventions. *Shibire*[ts] animals show intense nephrocyte turnover in the live labeling assay despite exposure to a temperature of 31°C for 2 hr which inhibits function of the fly dynamin during that period (**E–F"**). In contrast, blocking raft-mediated endocytosis for 2 hr by cyclodextrin in control nephrocytes strongly diminishes nephrin turnover and a large amount of the live labeled antibody is retained (**G–H"**). This suggests that nephrin turnover depends on raft-mediated endocytosis that occurs independent from dynamin function. The diffuse intracellular signal

*Figure 6 continued on next page*

*Figure 6 continued*

from live labeling was similar to control (*Figure 5—figure supplement 2A*). (**I**) Quantitation of results from (**E–H″**) expressed as ratio of the fluorescence intensity between surface and subcortical region for individual cells (mean ± standard deviation, n=11–12 animals per genotype, p>0.05 for *shibire^ts*, and p<0.0001 for cyclodextrin treatment).

The online version of this article includes the following figure supplement(s) for figure 6:

**Figure supplement 1.** Validation and controls for *Shibire^ts*.

is sufficient to block nephrin turnover and alter filtration characteristics. The *flo2*-dependent nephrin turnover thus appears to be required specifically for cleansing of the nephrocyte filtration barrier to maintain its permeability (working model, *Figure 7K*).

Taken together, our data indicate how a stable yet dynamic architecture of the filtration barrier facilitates its amazing capabilities and delineates the mechanistic role of endocytosis. Selective routes of vesicular transport are required for maintenance: Canonical dynamin-dependent endocytosis prevents lateral diffusion of slit diaphragm proteins to restrict slit diaphragms to their proper location while flotillin2-dependent endocytosis in lipid rafts facilitates nephrin turnover likely to promote dynamic flexibility but also to cleanse the barrier to prevent clogging during ceaseless filtration.

## Discussion

Here, we studied the mechanisms of slit diaphragm maintenance and the underlying role of endocytosis in *Drosophila* nephrocytes. Performing experiments that are currently precluded in mammalian or in vitro models, we combine knock-in lines into the genomic locus of nephrin with live imaging and short-term inhibition of endocytic functions. We observed a stable yet highly dynamic architecture that can be rebuilt after transient disruption. Although nephrin exhibited an extensive half-life exceeding 1 day, live antibody labeling and FRAP analysis suggested a rapid turnover of nephrin within minutes suggesting rapid cycles of uptake and recycling. To our knowledge, this is the first analysis of slit diaphragms dynamics in vivo. Upon acute silencing of *Rab5,* which impairs endocytic removal of ectopic nephrin, slit diaphragm proteins diffused laterally into the labyrinthine channels causing eventual breakdown of the architecture. At the same time, the size cut-off of the filtration barrier decreased suggesting incipient filter clogging. Acute disruption of dynamin function and cholesterol depletion revealed that endocytosis is required for two major functions that are attained by selective and independent transport routes: removal of ectopic nephrin by dynamin-dependent endocytosis and turnover of nephrin within the slit diaphragm by raft-mediated endocytosis. In this manner, endocytosis restricts and preserves the architecture and cleanses nephrin to preserve permeability of the filtration barrier. We identified *flotillin2* as a novel key protein in the raft-mediated turnover of nephrin.

The slit diaphragm is passed by vast amounts of plasma containing a wide range of proteins, metabolites, and xenobiotics. Binding of molecules to the slit diaphragm poses a constant threat of filter clogging. It has been a longstanding question how podocytes prevent clogging of the glomerular filter. Our live antibody experiments suggest that nephrin may shed proteins within the lower pH of endosomes – even antibodies binding with high affinity. We propose that this endosomal cleaning and rapid recycling of nephrin contributes to prevent filter clogging.

The exact speed of turnover is difficult to define. Antibody binding itself might speed up the endocytic turnover in our live antibody labeling assay or conversely impair endocytosis. The half-life of ~1 hr suggested by live labeling exceeds the half-life indicated by FRAP analysis, suggesting the latter. FRAP analysis might overestimate the speed of turnover since directly after photobleaching only bleached nephrin protein is subject to endocytosis while exclusively fluorescent nephrin is delivered by recycling. With progressive observation time, a steady state in the uptake and recycling of bleached and unbleached nephrin likely results in a premature plateau phase. Lateral diffusion of unbleached protein will further falsely diminish the half-life based on FRAP. Finally, the C-terminal tag might alter the kinetics of endocytosis. Thus, the rate of turnover can only be defined within a range of 7–60 min, which is not unlike the turnover that was described for adherens junctions (*de Beco et al., 2009*).

The filtration barrier in *Drosophila* nephrocytes differs anatomically from humans. Nevertheless, the functional and molecular correspondence is striking. The opportunities of genetic manipulation and accessibility for imaging of this podocyte model facilitated unique insights into the fundamental

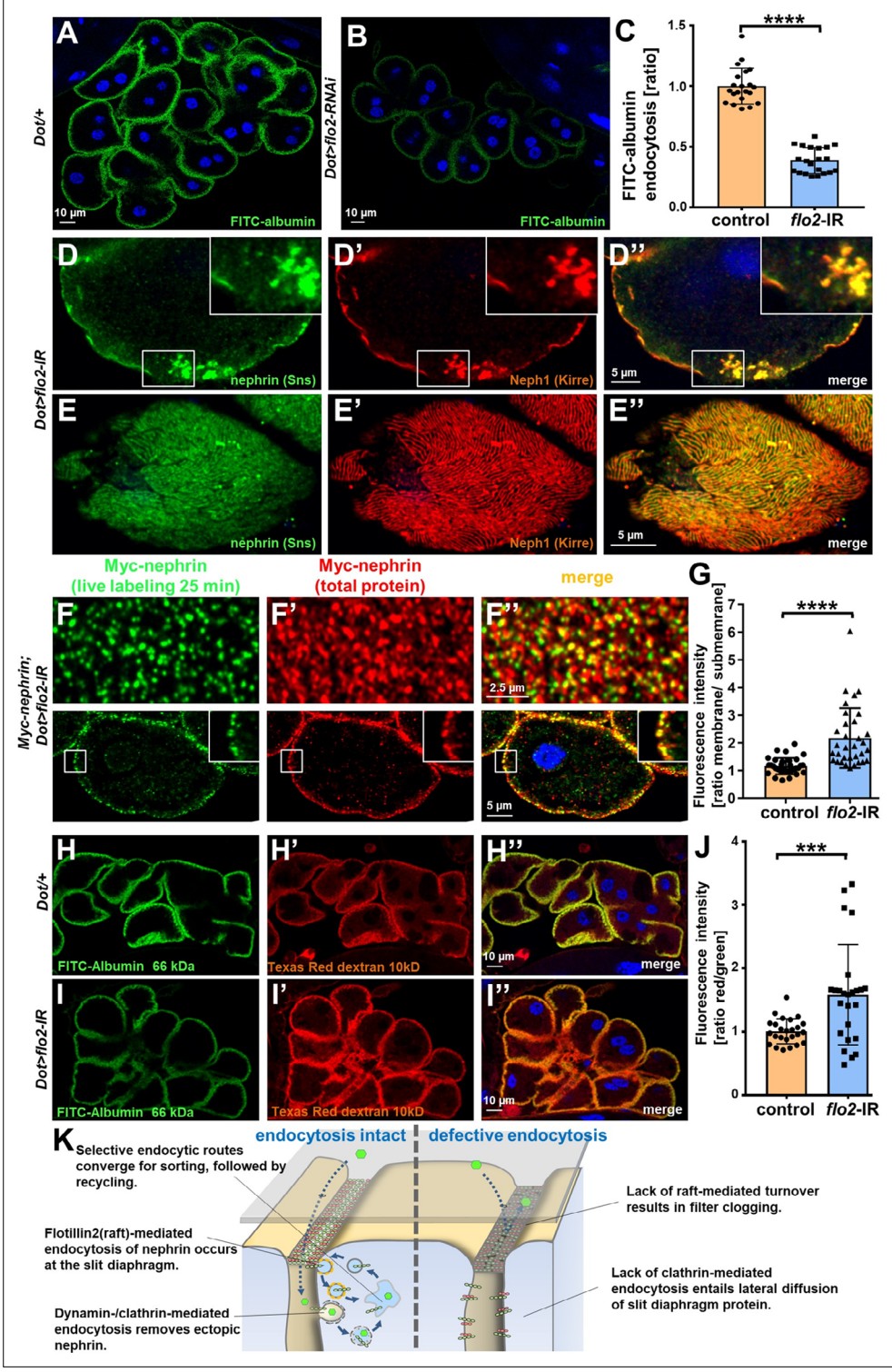

**Figure 7.** Flotillin2-mediated endocytosis is required for nephrin turnover in *Drosophila* nephrocytes. (**A–B**) Confocal microscopy images of nephrocytes after uptake of FITC-albumin as read-out of nephrocyte function are shown. Control nephrocytes exhibit stronger uptake (**A**) than nephrocytes expressing *flo2*-RNAi (**B**). (**C**) Quantitation of results analogous to (**A–B**) in ratio to a control experiment performed in parallel (mean ± standard deviation, n=7 animals per genotype, p<0.0001 for *flo2*-RNAi). (**D–E"**) Confocal images of nephrocytes expressing *flo2*-RNAi show localized breakdown of slit diaphragms in cross-sectional (**D–D"**) and tangential planes (**E–E"**). (**F–F"**) Confocal microscopy images in tangential sections (upper row) and cross sections (lower row) of nephrocytes

*Figure 7 continued on next page*

*Figure 7 continued*

are shown after live antibody labeling with 2 hr of chasing. Animals express flo2-RNAi under control of *Dorothy-GAL4*. Nephrin turnover is strongly reduced compared to control (*Figure 5—figure supplement 2A*). The diffuse intracellular signal from live labeling was similar to control (*Figure 5—figure supplement 2A*). (**G**) Quantitation of results from (**F**) compared to control experiments. Results are expressed as ratio of the fluorescence intensity between surface and subcortical regions for individual cells (mean ± standard deviation, n=11 animals per genotype, p<0.0001 for *flo2*-RNAi). (**H–I″**) Confocal microscopy images of nephrocytes after simultaneous uptake of FITC-albumin (66 kDa, green) and Texas-Red-Dextran (10 kDa) are shown. Control nephrocytes show significant uptake of both tracers (**H–H″**). Silencing of *flo2* causes a stronger decrease in the uptake of the larger tracer FITC-albumin compared to smaller Texas-Red-Dextran (**I–I″**). (**J**) Quantitation of fluorescence intensity expressed as a ratio of Texas-Red-Dextran/FITC-albumin (small/large tracer) confirms a disproportionate reduction for *flo2*-RNAi (mean ± standard deviation, n=9 animals per genotype, p<0.001 for *flo2*-RNAi). (**K**) Schematic illustrating the proposed mechanistic role of endocytosis for maintenance of the filtration barrier. Left: Ectopic fly nephrin within the channels is removed by clathrin-dependent endocytosis that returns most of the protein to the surface through recycling pathways. The nephrin that is bound within the slit diaphragm complex is subject to turnover in a shorter circuit that is raft-mediated and feeds into recycling as well. Right: Upon disruption of endocytosis filtration is impaired by clogging of the filter due to lack of cleansing and the architecture of the slit diaphragms is disturbed by unhindered lateral diffusion of slit diaphragm protein.

The online version of this article includes the following figure supplement(s) for figure 7:

**Figure supplement 1.** Silencing flotillin2 using a second RNAi line confirms reduced FITC-albumin uptake and altered permeability of the filtration barrier.

principles of filtration barrier maintenance in vivo. Selective cycles of endocytosis sustain a stable yet flexible filtration barrier. These basic principles are probably conserved in evolution. Since double knock-out mice of *Flotillin 1/Flotillin 2* were described without overt renal phenotype (*Bitsikas et al., 2014*), the exact molecular machinery may show partial divergence. It is conceivable that caveolins may be able to compensate the loss of flotillins in mammals. Future work in higher model organisms will be required to evaluate these principles in the mammalian kidney. Our data further support that mutations of the disease genes *GAPVD1* and *TBC1D8B* (*Hermle et al., 2018Hermle et al., 2018*; *Dorval et al., 2019*, *Kampf et al., 2019*) cause nephrotic syndrome via impaired endocytic trafficking. It will be important to clarify their specific roles in more detail.

Understanding the mechanistic role of endocytosis will help to identify novel angles for manipulation of the glomerular filtration barrier. Targeting the specific transport processes of nephrin is well suited to become a promising therapeutic strategy that may be effective across a wide range of glomerular diseases.

# Materials and methods

**Key resources table**

| Reagent type (species) or resource | Designation | Source or reference | Identifiers | Additional information |
|---|---|---|---|---|
| Gene (*Drosophila melanogaster*) | Nephrin (Sns) | Flybase | FLYB: FBgn0024189 | For simplicity we use the human name |
| Gene (*Drosophila melanogaster*) | Neph1 (Kirre) | Flybase | FLYB: FBgn0028369 | For simplicity we use the human name |
| Gene (*Drosophila melanogaster*) | Rab5 | Flybase | FLYB: FBgn0014010 | |
| Gene (*Drosophila melanogaster*) | Rab7 | Flybase | FLYB: FBgn0015795 | |
| Gene (*Drosophila melanogaster*) | Rab11 | Flybase | FLYB: FBgn0015790 | Transgenic animals |
| Gene (*Drosophila melanogaster*) | Shibire (Shi) | Flybase | FLYB: FBgn0003392 | Transgenic animals |
| Gene (*Drosophila melanogaster*) | Flotillin 2 (Flo2) | Flybase | FLYB: FBgn0264078 | Transgenic animals |
| Strain, strain background (*Drosophila melanogaster*) | Nephrin-RNAi (Sns-RNAi) | VDRC | VDRC #109442 | Transgenic animals |

*Continued on next page*

*Continued*

| Reagent type (species) or resource | Designation | Source or reference | Identifiers | Additional information |
|---|---|---|---|---|
| Strain, strain background (*Drosophila melanogaster*) | Nephrin-RNAi-2 (Sns-RNAi-2) | BDSC | BDSC #64872 | Transgenic animals |
| Strain, strain background (*Drosophila melanogaster*) | UAS-*Rab5*-RNAi | BDSC | BDSC #34832 | Transgenic animals |
| Strain, strain background (*Drosophila melanogaster*) | UAS-*Rab5*$^{S43N}$ | BDSC | BDSC #42704 | Transgenic animals, dominant negative variant |
| Strain, strain background (*Drosophila melanogaster*) | UAS-YFP-Rab5$^{Q88L}$ | BDSC | BDSC #9774 | Transgenic animals, constitutively active variant |
| Strain, strain background (*Drosophila melanogaster*) | UAS-*Rab7*-RNAi | BDSC | BDSC #27051 | Transgenic animals |
| Strain, strain background (*Drosophila melanogaster*) | UAS-YFP-*Rab7*$^{T22N}$ | BDSC | BDSC #9778 | Transgenic animals, dominant negative variant |
| Strain, strain background (*Drosophila melanogaster*) | UAS-*Rab11*-RNAi | BDSC | BDSC #42709 | Transgenic animals |
| Strain, strain background (*Drosophila melanogaster*) | UAS-*flo2*-RNAi | BDSC | BDSC #40833 | Transgenic animals |
| Strain, strain background (*Drosophila melanogaster*) | UAS-*flo2*-RNAi-2 | VDRC | VDRC #330316 | Transgenic animals |
| Strain, strain background (*Drosophila melanogaster*) | *Shibire*$^{ts}$ | BDSC | BDSC #2248 | Temperature-sensitive allele |
| Strain, strain background (*Drosophila melanogaster*) | Dorothy-*GAL4* | BDSC | BDSC #6903 | Transgenic, GAL4-dependent expression in nephrocytes |
| Strain, strain background (*Drosophila melanogaster*) | prospero-*GAL4* | ***Weavers et al., 2009*** (PubMed-ID: 18971929) | Promoter derived from: FLYB: FBgn0004595 | Transgenic, GAL4-dependent expression in nephrocytes |
| Strain, strain background (*Drosophila melanogaster*) | UAS-GFP-RNAi | BDSC | BDSC #41553 | Transgenic animals, control-RNAi |
| Strain, strain background (*Drosophila melanogaster*) | nephrin-GFP | This work | Edited gene: FLYB: FBgn0024189 | Insertion of GFP into the c-terminus of *sns,* (genomic) |
| Strain, strain background (*Drosophila melanogaster*) | Myc-nephrin | This work | Edited gene: FLYB: FBgn0024189 | Insertion of Myc into exon 2 of *sns* (genomic) |
| Antibody | anti-Sns (nephrin, rabbit polyclonal) | ***Bour et al., 2000*** (PubMed-ID: 10859168) | Target: FLYB: FBgn0024189 | 1:300 for IF |
| Antibody | anti-Kirre (Neph1, guinea pig, polyclonal) | ***Galletta et al., 2004*** (PubMed-ID: 15511638) | Target: FLYB: FBgn0028369 | 1:200 for IF |
| Antibody | anti-Rab5 (rabbit, polyclonal) | Abcam | ab18211 | 1:200 for IF |
| Antibody | anti-Rab7 (mouse, monoclonal) | DSHB | Rab7 | 1:100 for IF |
| Antibody | anti-Myc (mouse, monoclonal) | DSHB | 9E10 | 1:100 for IF |
| Antibody | anti-Myc (mouse, monoclonal) | Santa Cruz Biotechnology | sc-40 | 1:100 for IF |
| Antibody | anti-RAB11 (rabbit, monoclonal) | Cell Signaling Technology | 5589S | 1:100 for IF |
| Antibody | Alexa Fluor 488 anti-rabbit, (donkey, polyclonal) | Thermofisher | #A-21206 | 1:200 for IF, secondary antibody |

*Continued on next page*

*Continued*

| Reagent type (species) or resource | Designation | Source or reference | Identifiers | Additional information |
|---|---|---|---|---|
| Antibody | Alexa Fluor 488 anti-mouse (donkey, polyclonal) | Thermofisher | #A32766 | 1:200 for IF, secondary antibody |
| Antibody | Alexa Fluor 568 anti-rabbit (donkey, polyclonal) | Thermofisher | #A10042 | 1:200 for IF, secondary antibody |
| Antibody | Alexa Fluor 568 anti-mouse (donkey, polyclonal) | Thermofisher | #A10037 | 1:200 for IF, secondary antibody |
| Antibody | Alexa Fluor 568 anti-guinea pig (goat, polyclonal) | Thermofisher | #11075 | 1:200 for IF, secondary antibody |
| Commercial assay or kit | In Situ Cell Death Detection Kit | Sigma/Roche | 11684795910 | TUNEL labeling |
| Chemical compound, drug | FITC-albumin | Sigma/Merck | A9771 | Final conc.: 0.2 mg/ml |
| Chemical compound, drug | Texas-Red-Dextran | Thermofisher | D1863 | Final conc.: 0.2 mg/ml |
| Chemical compound, drug | Texas-Red-Avidin | Thermofisher | A2348 | Final conc.: 0.2 mg/ml |
| Chemical compound, drug | Alexa 488 wheat germ agglutinin | Thermofisher | W11261 | Final conc.: 0.2 mg/ml |
| Chemical compound | Roti-Mount | Carl Roth | HP19.1 | For mounting |
| Chemical compound, drug | Hoechst 33342 | Thermofisher | H1399 | 1:1000 for IF |
| Chemical compound, drug | Bafilomycin | Invivogen | tlrl-baf1 | Final conc.: 0.1 µM |
| Chemical compound, drug | Methyl-β-cyclodextrin | Sigma-Aldrich/Merck | 332615 | Final conc.: 10 mM |
| Chemical compound,drug | Low-melting-agarose | Carl Roth | #6351.5 | Final use: 1% agarose |
| Chemical compound, drug | Schneider's insect medium | Sigma-Aldrich/Merck | #S0146 | |
| Chemical compound, drug | Triton X-100 | Sigma-Aldrich/Merck | #9036-19-5 | Final conc.: 0.1% in PBS |
| Software, algorithm | GraphPad Prism | GraphPad Inc | GraphPad Prism 9.3.1 | |
| Software, algorithm | Fiji/ImageJ | Open source | ImageJ 2.1.0/1.53c | |
| Software, algorithm | GIMP | Open source | GIMP 2.10 | |

## Fly strains and husbandry

Flies were reared on standard food at room temperature, 18°C, 25°C, or 31°C as indicated. Overexpression and transgenic RNAi studies were performed using the UAS/*GAL4* system (RNAi crosses grown at 25°C or 31°C). Nephrocyte indicates the subtype of garland cell nephrocytes throughout the manuscript. Stocks obtained from the Bloomington *Drosophila* Stock Center (BDSC) were UAS-*nephrin*(sns)-RNAi (#64872), UAS-*Rab5*-RNAi (#34832), UAS-*Rab5^S43N* (dominant negative) (#42704), UAS-YFP-Rab5^Q88L (constitutively active, #9774), UAS-*Rab7*-RNAi (#27051), UAS-YFP-*Rab7^T22N* (dominant negative #9778), UAS-*Rab11*-RNAi (#42709), UAS-*flo2*-RNAi (#40833), and *Shibire^ts* (#2248). The second UAS-*flo2*-RNAi (#330316) and UAS-*nephrin*(sns)-RNAi-2 (VDRC #109442) were provided by the Vienna *Drosophila* RNAi Center (VDRC), *prospero-GAL4* (**Weavers et al., 2009**) and *Dorothy-GAL4* (#6903; BDSC) were used with or without *tub-GAL80^ts* (#7018; BDSC) to control expression in nephrocytes. UAS-GFP-RNAi (#41553; BDSC) or wild-type (yw^1118) were crossed to *GAL4*-drivers as control.

## Generation of nephrin-GFP

Nephrin-GFP was generated by using CRISPR/piggyBac to introduce a C-terminal super folder GFP at the fly nephrin (*sns*) locus using the scarless gene editing approach (*Bruckner et al., 2017*). A single guide RNA targeting the 3' end of *sns* was cloned into pU6-BbsI-chiRNA. A dsDNA donor template for homology-directed repair with 1 kb homologies upstream and downstream was generated by PCR amplification from genomic DNA and assembly into pScarlessHD-sfGFP-DsRed by Gibson DNA Assembly (New England Biolabs). A mixture of both plasmids was injected into flies expressing Cas9 under *nos* regulatory sequences by BestGene. CRISPR-edited lines were identified by the presence of DsRed eye fluorescence. We removed PBac-3xP3-DsRed-PBac sequences in these stocks by precise excision of the PBac transposable element by crossing to tub-Pbac flies (#8283; BDSC) and established the resulting nephrin-GFP as homozygous stocks.

## Live imaging using nephrin-GFP

Nephrin-GFP expressing nephrocytes were dissected in phosphate buffered saline (PBS) immediately before mounting on slides with cover slips in Schneider's medium (#S0146, Sigma-Aldrich/Merck) containing 1% low melting agarose (#6351.5, Carl Roth GmbH). The slide was put on an ice block for a few seconds and then left on room temperature for 5 min to allow the agarose solution to become solid. Imaging was performed using a Zeiss LSM 880 laser scanning microscope employing electronic autofocus over the course of up to 1 hr.

## Generation of genomic Myc-nephrin

For generation of genomic Myc-nephrin, we targeted the second exon of sns using using pCFD3 (#49410; Addgene, target sequence: AGTGCCAGGTGGGACCGGCT). A homology-directed repair template was assembled by a step-wise amplification of homologies upstream and downstream of the second exon of fly nephrin (*sns*) using a vector from the BACPAC library that covered the *sns* locus. A Myc sequence was inserted directly adjacent to the target's (mutated) PAM. DsRed cDNA under P3 promoter flanked by loxP sites was derived from pHD-DsRed (Addgene plasmid #51434) and placed into the flanking intron that preceded the downstream homology. Twelve synonymous changes were introduced between Myc and the exon boundary to avoid alignment in the interjacent section. A mixture of both plasmids was injected into flies expressing Cas9 under nos regulatory sequences (#54591; BDSC) by BestGene. CRISPR-edited lines were identified by the presence of DsRed eye fluorescence and the DsRed marker was removed by crossing to flies expressing cre recombinase (#1092; BDSC). We established the resulting Myc-nephrin flies as a homozygous stock.

## Fluorescent tracer uptake

Fluorescent tracer uptake in nephrocytes to evaluate nephrocyte function was performed as previously described (*Hermle et al., 2017*). Briefly, nephrocytes were dissected in PBS and incubated with FITC-albumin (#A9771, Sigma) for 30 s. After a fixation step of 5 min in 8% paraformaldehyde, cells were rinsed in PBS and exposed to Hoechst 33342 (1:1000, #H1399, Thermofisher) for 20 s and mounted in Roti-Mount (#HP19.1, Carl Roth). Cells were imaged using a Zeiss LSM 880 laser scanning microscope. Quantitation of fluorescent tracer uptake was performed with ImageJ software. The results are expressed as a ratio to a control experiment with flies carrying the (heterozygous) *GAL4* transgene but no UAS that was performed in parallel.

The parallel recording of two fluorescent tracers of different size to study the passage of the slit diaphragm was carried out in the same way as the assay for nephrocyte function, except that nephrocytes were simultaneously incubated with tracers FITC-albumin (0.2 mg/ml) and Texas-Red-Dextran (#D1863, Thermofisher, 10 kDa, 0.2 mg/ml) for 30 s after dissection. Cells were imaged using a Zeiss LSM 880 laser scanning microscope. Image quantitation was performed with ImageJ software for each channel separately. Alternative tracers were Texas-Red-Avidin (66 kDa, #A2348, Sigma) and Alexa Fluor 488 wheat germ agglutinin (38 kDa, #W11261, Thermofisher).

## Immunofluorescence studies and TUNEL detection using *Drosophila* tissue

For immunofluorescence, nephrocytes were dissected, fixed for 20 min in PBS containing 4% paraformaldehyde, and stained according to the standard procedure. The following primary antibodies were

used: rabbit anti-sns (*Bour et al., 2000*) (1:300, gift from S Abmayr) and guinea pig anti-Kirre (*Galletta et al., 2004*) (1:200, gift from S Abmayr). Other antibodies used were rabbit anti-Rab5 (ab18211, abcam, 1:100), mouse anti-Rab7 (Rab7, DSHB 1:100), mouse anti-Myc (9E10; DSHB, 1:100), mouse anti-c-Myc (sc-40; Santa Cruz Biotechnology 1:100), and rabbit anti-RAB11 (#5589S; Cell Signaling Technology, 1:100). The following secondary antibodies were used: Alexa Fluor 488 donkey anti-rabbit (#A-21206, Thermofisher, 1:200), Alexa Fluor 488 donkey anti-mouse (#A32766, Thermofisher, 1:200), Alexa Fluor 568 donkey anti-rabbit (#A10042, Thermofisher, 1:200), Alexa Fluor 568 donkey anti-mouse (#A10037, Thermofisher, 1:200), Alexa Fluor 568 goat anti-guinea pig (#11075, Thermofisher, 1:200).

Apoptotic cells were visualized using the In Situ Cell Death Detection Kit (#11684795910, Sigma/Roche) according to the manufacturer's instructions. For imaging, a Zeiss LSM 880 laser scanning microscope was used. Image processing was done by ImageJ and GIMP software.

## Live antibody labeling and internalization

For live antibody labeling, we modified previously published protocols (*Strutt et al., 2011*; *Hermle et al., 2013*). Nephrocytes were dissected in PBS and immediately incubated with primary antibody (mouse anti-c-Myc, 9E10; DSHB, 1:100 in PBS) for 25 min at 4°C before rinsing four times with cold PBS to remove unbound antibody. The living cells labeled with primary anti-Myc antibody were chased at 29°C in Schneider's insect medium for the indicated time. For lipid raft inhibition chase was performed with 10 mM methyl-β-cyclodextrin (#332615, Sigma-Aldrich/Merck) diluted in Schneider's insect medium and for inhibiton of endosomal acidification chase was performed with 0.1 μM bafilomycin (#tlrl-baf1, Invivogen) diluted in Schneider's insect medium. Then, the tissue was fixed in PBS containing 4% paraformaldehyde for 20 min, permeabilized using PBS containing 0.1% Triton X-100 (#T8787, Sigma-Aldrich/Merck), and washed briefly three times before Alexa Fluor 488-coupled anti-mouse secondary antibody was applied (#A32766, Thermofisher) for 2 hr at room temperature. To obtain total nephrin staining after this step, incubation with mouse anti-Myc primary antibody (sc-40; Santa Cruz Biotechnology 1:100) was repeated overnight after washing. After the preceding permeabilization, the entire nephrin protein of the cell was now accessible to the anti-Myc antibody in this step. Finally, for detection of total nephrin staining an Alexa Fluor-568-coupled anti-mouse secondary (#A10037, Thermofisher) was applied for 2 hr at room temperature. For imaging, a Zeiss LSM 880 laser scanning microscope was utilized. Image processing was done by ImageJ and GIMP software.

## FRAP analysis

Nephrocytes from wandering third instar expressing nephrin-GFP were dissected and mounted in Schneider's media. FRAP experiments were conducted on a Zeiss 880 confocal microscope. Pre-bleach images (two to four frames) were first acquired, followed by a single photobleaching event consisting of 30–40 scans of the 488 nm laser at 100% power. Photobleaching was confined to a region of interest (ROI) covering a small region of slit diaphragms, as indicated by enriched nephrin-GFP at the cell surface. After bleaching, standard time series acquisition (images acquired every 10–15 s) continued for the remainder of the movie. To counter any sample drift, manual correction of the z-axis was performed throughout the time series acquisition. An ROI was drawn over the slit diaphragm-containing edge of the cell within the photobleached area using ImageJ. We measured mean gray value in the ROI at each time point, and subtracted the background from an adjacent extracellular region outside the ROI. For each cell, we measured one to two ROIs. We calculated the mean of the mean gray values for the pre-bleach period, and then standardized all subsequent measures of signal intensity by expressing them as a percentage of the pre-bleach signal. We then averaged the percent mean gray values for all acquisition time points within each minute of the post-bleach time lapse series. We combined these data from all the FRAP experiments, treating each ROI from each cell as a replicate, to determine the mean and SEM at each time point.

## Channel diffusion assay

To visualize the nephrocytes' membrane invaginations, we dissected nephrocytes and fixed them briefly for 5 min in PBS containing 4% paraformaldehyde (#15700, Electron Microscopy Sciences). Shorter fixation preserves slit diaphragm permeability. Cells were then incubated for 10 min in FITC-albumin (Sigma) or Texas-Red-Dextran (10 kD; Thermofisher) to allow tracer diffusion into the channels.

Our regular staining protocol was completed according to standard procedure after a second fixation step in paraformaldehyde for 15 min.

## Electron microscopy

For transmission electron microscopy (TEM) nephrocytes were dissected and fixed in 4% formaldehyde and 0.5% glutaraldehyde in 0.1 M cacodylate buffer, pH 7.4 (EM facility, Harvard Medical School). TEM was carried out using standard techniques.

## Statistics

Paired t-test was used to determine the statistical significance between two interventions. One-way ANOVA followed by Dunnett's correction for multiple testing (unless otherwise indicated) was used for multiple comparisons (GraphPad Prism software). Measurements were from distinct samples. Asterisks indicate significance as follows: *$p < 0.05$, **$p < 0.01$, ***$p < 0.001$, ****$p < 0.0001$. A statistically significant difference was defined as $p < 0.05$. Error bars indicate standard deviation (SD). At least three repetitions were performed per experiments with a number of animals suitable to the approach ranging from 1 to 5. This results in the number of N ranging from 5 to 14. No specific a priori calculation of sample size was performed. No data or outliers were excluded.

## Acknowledgements

We thank C Meyer for excellent technical support and R Nitschke, Life Imaging Centre, University of Freiburg, for help with confocal microscopy. We thank the Developmental Studies Hybridoma Bank (DSHB) for antibodies and the Bloomington *Drosophila* Stock Center and Vienna *Drosophila* Resource Center for providing fly stocks. This research was supported by grants from the Deutsche Forschungsgemeinschaft (DFG) to TH (HE 7456/3-1, HE 7456/4-1), and project-ID 431984000 – SFB 1453 (to TH, MK, and GW), TRR 152 (to MK) project-ID 239283807, and Germany's Excellence Strategy: CIBSS – EXC-2189 – project-ID 390939984 (to GW and MK). JM and HH were supported by the MOTI-VATE program of the Medical Faculty of the University of Freiburg. MC was supported by the China Scholarship Council. TH was supported by the Berta-Ottenstein-Programme for Advanced Clinician Scientists, Faculty of Medicine, University of Freiburg. TH and KL acknowledge support from the Deutsche Gesellschaft für Innere Medizin (DGIM). JSP and AS were supported by UNC Kidney Center endowment funds.

## Additional information

### Funding

| Funder | Grant reference number | Author |
| --- | --- | --- |
| Deutsche Forschungsgemeinschaft | project-ID 431984000 – SFB 1453 | Tobias Hermle Michael Köttgen Gerd Walz |
| Deutsche Forschungsgemeinschaft | HE 7456/3-1 | Tobias Hermle |
| Deutsche Forschungsgemeinschaft | HE 7456/4-1 | Tobias Hermle |
| Deutsche Forschungsgemeinschaft | TRR 152 – project-ID 239283807 | Michael Köttgen |
| Germany's Excellence Strategy | CIBSS – EXC-2189 – project-ID 390939984 | Gerd Walz Michael Köttgen |
| University of Freiburg | MOTI-VATE program of the Medical Faculty | Julian Milosavljevic Helena Heinkele |
| China Scholarship Council | | Mengmeng Chen |

| Funder | Grant reference number | Author |
|---|---|---|
| University of Freiburg | Berta-Ottenstein-Programme for Advanced Clinician Scientists | Tobias Hermle |
| Deutsche Gesellschaft für Innere Medizin | Clinician Scientist Fellowship | Tobias Hermle Konrad Lang |
| UNC Kidney Center endowment funds. | | John Poulton Andrew Spracklen |

The funders had no role in study design, data collection and interpretation, or the decision to submit the work for publication.

## Author contributions

Konrad Lang, Formal analysis, Investigation, Writing - review and editing; Julian Milosavljevic, Helena Heinkele, Mengmeng Chen, Lea Gerstner, Dominik Spitz, Severine Kayser, Martin Helmstädter, Andrew Spracklen, Investigation; Gerd Walz, Resources, Writing - review and editing; Michael Köttgen, Conceptualization, Writing - review and editing; John Poulton, Investigation, Writing - review and editing; Tobias Hermle, Conceptualization, Formal analysis, Supervision, Funding acquisition, Investigation, Writing - original draft, Project administration, Writing - review and editing

## Author ORCIDs

Michael Köttgen http://orcid.org/0000-0003-2406-5039
Andrew Spracklen http://orcid.org/0000-0002-5550-8595
Tobias Hermle http://orcid.org/0000-0002-0441-7749

## Decision letter and Author response

Decision letter https://doi.org/10.7554/eLife.79037.sa1
Author response https://doi.org/10.7554/eLife.79037.sa2

# Additional files

## Supplementary files

- MDAR checklist
- Reporting standard 1. ARRIVE Essential 10.

## Data availability

Transgenic *Drosophila* lines are available from the corresponding author upon reasonable request. Unprocessed image files were submitted to a public repository (zenodo.org, DOI: https://doi.org/10.5281/zenodo.6418762). Access is not restricted for scientific purposes.

The following dataset was generated:

| Author(s) | Year | Dataset title | Dataset URL | Database and Identifier |
|---|---|---|---|---|
| Lang K, Hermle T | 2022 | Selective endocytosis controls slit diaphragm maintenance and dynamics - Supplementary Dataset | https://doi.org/10.5281/zenodo.6418762 | Zenodo, 10.5281/zenodo.6418762 |

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
