## [Editor Report]

This article would be of interest to all researchers who work in understanding the mechanisms involved in podocyte slit diaphragm homeostasis and maintenance of the glomerular filtration barrier. The work provides substantial new insights into nephrin dynamics and the mechanisms of slit diaphragm maintenance. A series of compelling experiments depicted that dynamin-mediated endocytosis was involved in ectopic nephrin turnover and that flotillin-mediated turnover of nephrin occurred within the slit diaphragm was needed to maintain filter permeability in-vivo.

---

## [Decision Letter]

**Decision letter after peer review:**

Thank you for submitting your article "Selective endocytosis controls slit diaphragm maintenance and dynamics in *Drosophila* nephrocytes" for consideration by *eLife*. Your article has been reviewed by 3 peer reviewers, including Ilse S Daehn as the Reviewing Editor and Reviewer #1, and the evaluation has been overseen by Mone Zaidi as the Senior Editor. The following individual involved in the review of your submission has agreed to reveal their identity: Gábor Juhász (Reviewer #3).

Essential revisions:

1) The current version of the presented data would benefit from more convincing data to prove endocytosis in the pulse-chase experiments. Reviewer #2 would like to see better myc-labeled nephrin in endocytotic vesicles and a demonstration of a stable pulse-chase method.

2) More discussion regarding what is known from the human genetic studies and the relevance of endocytic pathways described here in relation to mistrafficking of nephrin.

*Reviewer #1 (Recommendations for the authors):*

The human relevance focus on endocytic pathways is suggested in the introduction section, work previously performed by the group. However, the authors do not attempt to address this in their system or discuss it in the Discussion section. Given that the authors have previously described patient-derived mutations in nephrotic syndrome such as GAPVD1, and that silencing Gapvd1 in *Drosophila* impaired endocytosis and caused mistrafficking of nephrin, the question of endocytic pathways and nephrin turnover implicated in patients with GAPVD1 mutations could be addressed. Please also elaborate further in the discussion.

*Reviewer #2 (Recommendations for the authors):*

The authors present their work clearly and with well-prepared figures.

Figure 1 – The myc-tag pulse-chase experiments show a myc label at the cell surface but fails to be detected after cell surface disappearance. Why is there no submembranous vesicular pattern? The statement about "diffuse intracellular signal" is not convincing and should not be linked with endocytosis as no vesicular pattern is shown here. An experimental model with a sustained nephrin label after leaving the cell membrane would strengthen a lot the made statement about endocytosis.

Figure 2 – The authors investigate and quantify the nephrin cell surface dynamics by 1. myc-nephrin pulse label – chase approach for endocytosis (G) and 2. nephrin-gfp photobleaching for cell surface (re-)appearance (I). How could you explain the different half times (T_1/2_): ~2h for nephrin endocytosis and 7 min for nephrin cell surface reappearance? How can this imbalance of stronger nephrin cell surface appearance be compensated by the nephrocyte?

Leaving the reader puzzled are Figures 6F, H, and 7I, where myc-labeled nephrin is observed in presumably endocytotic vesicles. Why can these vesicles be observed here but not in the experiments shown in Figure 1 (s. above)? How stable is this pulse-chase method? Please provide more data and information on this.

Throughout the manuscript, the method of fluorescent intensity (ratio membrane/submembrane) is used. Please explain/describe in detail how this is done and what is used for calibration?

The Video files are of good technical quality. A scale for dimension and time would help the observer a lot.

---

## [Author Response]

Essential revisions:1) The current version of the presented data would benefit from more convincing data to prove endocytosis in the pulse-chase experiments. Reviewer #2 would like to see better myc-labeled nephrin in endocytotic vesicles and a demonstration of a stable pulse-chase method.

Thank you very much for careful review and the helpful comments! We understand the reviewers' concerns since the antibody is removed surprisingly fast upon endocytosis leaving only a faint signal detectable by anti-Myc staining. However, large endocytic vesicles containing nephrin were not detectable in live cell imaging (Video 2), suggesting that transport occurs rapidly within smaller vesicles. Furthermore, we would like to point out that several lines of evidence validate endocytosis in the live antibody labeling assay. Silencing of *Rab5* causes extensive retention of the antibody on the cell surface, clearly suggesting that endocytosis is required to remove the antibody from the slit diaphragm. Similarly, blocking of raft-mediated endocytosis either by silencing of *flotillin 2* or dispersal of cholesterol causes retention of the antibody. While the antibody is not detectable in endocytic vesicles under control conditions, this changes entirely upon silencing of Rab7. The impaired degradation slows down the process sufficiently to now detect vesicles containing antibody bound to nephrin but also dissociated from the protein. This verifies that the antibody is removed by endocytosis. To confirm this independently within the identical genetic background, we now performed additional experiments and exposed nephrocytes after live labeling to bafilomycin. This V-ATPase inhibitor blocks acidification of endosomes and slows down degradation in this manner. In turn, this treatment entailed the appearance of vesicles containing the live labeled antibody partially colocalizing with the total protein. We feel that the additional experimental data together with our previous findings firmly substantiate the occurrence of endocytosis.

We now refer to the new data as follows in the manuscript:

“Apparently, the live-labeled antibody was rapidly degraded, but we detected a vesicular signal when degradation was slowed by bafilomycin-mediated inhibition of endolysosomal acidification (Figure 2—figure supplement 1F-G).”

2) More discussion regarding what is known from the human genetic studies and the relevance of endocytic pathways described here in relation to mistrafficking of nephrin.

We completely agree with the reviewer that the human relevance of the role of endocytosis is rooted in the human genetic findings and we now take up this important aspect in the discussion again (p14, discussion):

“Our data further support that mutations of the disease genes *GAPVD1* and *TBC1D8B*(Hermle et al., 2018, Dorval et al., 2019, Kampf et al., 2019) cause nephrotic syndrome via impaired endocytic trafficking. It will be important to clarify their specific roles in more detail.”

Reviewer #1 (Recommendations for the authors):The human relevance focus on endocytic pathways is suggested in the introduction section, work previously performed by the group. However, the authors do not attempt to address this in their system or discuss it in the Discussion section. Given that the authors have previously described patient-derived mutations in nephrotic syndrome such as GAPVD1, and that silencing Gapvd1 in *Drosophila* impaired endocytosis and caused mistrafficking of nephrin, the question of endocytic pathways and nephrin turnover implicated in patients with GAPVD1 mutations could be addressed. Please also elaborate further in the discussion.

We apologize that this important aspect did not get sufficient attention in the discussion. As outlined above, we followed the reviewer’s reasonable advice and now include the following sentence in the discussion (p.14):

“Our data further support that mutations of the disease genes *GAPVD1* and *TBC1D8B*(Hermle et al., 2018, Dorval et al., 2019, Kampf et al., 2019) cause nephrotic syndrome via impaired endocytic trafficking. It will be important to clarify their specific roles in more detail.”

Exploring the functional role of these genes is part of our scientific efforts and a manuscript dealing with these questions is currently in revision elsewhere. However, since the manuscript submitted to *eLife* is entirely focused on the mechanistic role of endocytosis in general, it would be beyond its scope to include additional data regarding GAPVD1 or TBC1D8B. Furthermore, space restrictions would prevent us from doing that in a meaningful way.

Reviewer #2 (Recommendations for the authors):The authors present their work clearly and with well-prepared figures.Figure 1 – The myc-tag pulse-chase experiments show a myc label at the cell surface but fails to be detected after cell surface disappearance. Why is there no submembranous vesicular pattern? The statement about "diffuse intracellular signal" is not convincing and should not be linked with endocytosis as no vesicular pattern is shown here. An experimental model with a sustained nephrin label after leaving the cell membrane would strengthen a lot the made statement about endocytosis.

It is indeed surprising that the antibody is degraded so rapidly that no vesicular pattern is observed after live labeling under control conditions. However, as already outlined above, a vesicular staining pattern of the antibody becomes visible once the degradation is slowed down by *Rab7*-RNAi. This also indicates that endocytosis of the live labeled endocytosis does occur. We further performed additional experiments and now confirm this observation upon exposure of nephrocytes to bafilomycin during the chase period, which blocks endolysosomal acidification. Again, slower endolysosomal degradation allows detection of the live labeled antibody in a vesicular pattern. This corroborates our observations with Rab7-RNAi within a wild-type background. Furthermore, the live labeled antibody is strongly retained on the cell surface upon general inhibition of endocytosis by *Rab5*-RNAi or specifically after blocking raft-mediated endocytosis, which indicates that it cannot be removed properly upon defective endocytosis. Taken together, we feel our data demonstrates endocytosis of the live labeled antibody.

Figure 2 – The authors investigate and quantify the nephrin cell surface dynamics by 1. myc-nephrin pulse label – chase approach for endocytosis (G) and 2. nephrin-gfp photobleaching for cell surface (re-)appearance (I). How could you explain the different half times (T_1/2_): ~2h for nephrin endocytosis and 7 min for nephrin cell surface reappearance? How can this imbalance of stronger nephrin cell surface appearance be compensated by the nephrocyte?

Thank you for pointing this out, our discussion did not adequately address this discrepancy. The half-life of the majority of nephrin protein on the cell surface must be closer to 1 h since most of the signal is gone after 2 h. The half-life of the very small stable fraction cannot be assessed properly by this assay. Still, one hour appears longer indeed than the half-life suggested by the FRAP experiments that was in the range of 7 min. The true speed cannot determined reliably within the range from 7-60 min at this point. Endocytosis may be decelerated by antibody binding and the FRAP analysis is likely to overestimate the speed with several possible explanations. We included the following text in the discussion:

“The exact speed of turnover is difficult to define. Antibody binding itself might speed up the endocytic turnover in our live antibody labeling assay or conversely impair endocytosis. The half-life of ~1 h suggested by live labeling exceeds the half-life indicated by FRAP analysis is pointing towards the latter. FRAP analysis might overestimate the speed of turnover since directly after photobleaching only bleached nephrin protein is subject to endocytosis while exclusively fluorescent nephrin is delivered by recycling. With progressive observation time, a steady state in the uptake and recycling of bleached and unbleached nephrin likely results in a premature plateau phase. Lateral diffusion of unbleached protein will further falsely diminish the half-life based on FRAP. Finally, the C-terminal tag might alter the kinetics of endocytosis. Thus, the rate of turnover can only be defined within a range of 7-60 minutes, which is not unlike the turnover that was described for adherens junctions(de Beco et al., 2009).”

Leaving the reader puzzled are Figures 6F, H, and 7I, where myc-labeled nephrin is observed in presumably endocytotic vesicles. Why can these vesicles be observed here but not in the experiments shown in Figure 1 (s. above)? How stable is this pulse-chase method? Please provide more data and information on this.

The only cross-sectional image in Figure 1 shows live imaging using nephrin-GFP. In this divergent setting intracellular background signal was mostly lacking and vesicles were rarely detectable. Similarly, such a signal was absent after shorter chase periods (Figure 2D-E). However, in the correct control settings with a chase period of 2 h after live labeling in Figure 5—figure supplement 2A (heterozygous) or Figure 2F (homozygous) we similarly detected a diffuse intracellular signal including small dots. We observed no systematic difference of control and intervention regarding the faint and occasionally dotty intracellular signal in Figure 6F and H and 7F. For clarification, this information has now been included in the figure legends: “The diffuse intracellular signal from live labeling was similar to control (Figure 5—figure supplement 2A).” Figure 7I shows another experiment (FITC-albumin uptake). Generally, we find it difficult to assign the term vesicles to these very small dots, since a similar signal may occur in stainings (including those without live labeling).

Throughout the manuscript, the method of fluorescent intensity (ratio membrane/submembrane) is used. Please explain/describe in detail how this is done and what is used for calibration?

We quantified fluorescence within a narrow region of interest (ROI) that followed the cell membrane in an equatorial section for at least ¼ of the whole circumference of a cell using imageJ. It was ensured that the section of the cell membrane used for quantitation was not immediately adjacent to another cell. The average intensity derived from this measurement was divided by the average fluorescence intensity of a ROI in the immediately adjacent domain below the cell membrane. The depth of this subcortical ROI was defined as approximately the outer third of the distance from just below the surface to the center of the cell (see illustration Figure 2—figure supplement 1E). This region corresponds the area where most of the vesicular trafficking occurs and this approach mostly excludes the perinuclear domain including the ER where vast organelles dominate over vesicular transport. The fluorescence intensity derived from live labeling on the surface declines proportionally over time while the signal in the subcortical domain mildly increases. These changes reflect endocytic turnover. We consciously chose an internal control within the same cell since even with identical settings, there is a certain degree of variation in signal strength between images. This is likely due to a combination of inevitable technical variability in the procedures for staining and recording as well as inter individual variation. The internal control eliminates a large part of the technical variability.

The Video files are of good technical quality. A scale for dimension and time would help the observer a lot.

Thank you very much for reminding us, scale bars and a time stamp have been added to the Videos.